

# The incorporation of the Tripleclouds concept into the $\delta$-Eddington two-stream radiation scheme: solver characterization and its application to shallow cumulus clouds

Nina Črnivec[1] and Bernhard Mayer[1,2]

[1]Chair of Experimental Meteorology, Ludwig-Maximilians-Universität München, Munich, Germany
[2]Institut für Physik der Atmosphäre, Deutsches Zentrum für Luft- und Raumfahrt, Oberpfaffenhofen, Germany

**Correspondence:** Nina Črnivec (nina.crnivec@physik.uni-muenchen.de)

**Abstract.** The treatment of unresolved cloud-radiation interactions in weather and climate models has considerably improved over the recent years, compared to conventional plane-parallel radiation schemes, which previously persisted in these models for multiple decades. One such improvement is the state-of-the-art Triplecloud radiative solver, which has two cloudy and one cloud-free region in each vertical model layer and is thereby capable of representing cloud horizontal inhomogeneity. Inspired
by the Triplecloud concept, primarily introduced by Shonk and Hogan (2008), we incorporated a second cloudy region into the widely employed $\delta$-Eddington two-stream method with maximum-random overlap assumption for partial cloudiness. The inclusion of another cloudy region in the two-stream framework required an extension of vertical overlap rules. While retaining the maximum-random overlap for the entire layer cloudiness, we additionally assumed the maximum overlap of optically thicker cloudy regions in pairs of adjacent layers. This extended overlap formulation implicitly places the optically thicker
region towards the interior of the cloud, which is in agreement with the core-shell model for convective clouds. The method was initially applied on a shallow cumulus cloud field, evaluated against a three-dimensional benchmark radiation computation. Different approaches were used to generate a pair of cloud condensates characterizing the two cloudy regions, testing various condensate distribution assumptions along with global cloud variability estimate. Regardless of the exact condensate setup, the radiative bias in the vast majority of Triplecloud configurations was considerably reduced compared to the conventional
plane-parallel calculation. Whereas previous studies employing the Triplecloud concept focused on researching the top-of-the-atmosphere radiation budget, the present work pioneeringly applies the Triplecloud to atmospheric heating rate and net surface flux. The Triplecloud scheme was implemented in the comprehensive *libRadtran* radiative transfer package and can be utilized to further address key scientific issues related to unresolved cloud-radiation interplay in coarse-resolution atmospheric models.

## 1 Introduction

Radiation schemes in coarse-resolution numerical weather prediction and climate models, commonly referred to as general circulation models (GCMs), have traditionally been claimed to be impaired by the poor representation of clouds (Randall et al., 1984, 2003, 2007). Undoubtedly, one of the most rigorous assumptions that persisted in GCMs for multiple decades, was





the complete removal of cloud horizontal heterogeneity − the so-called plane-parallel cloud representation (Fig. 1, bottom

left). Since the nature of cloud-radiation interactions is intrinsically nonlinear, the plane-parallel representation of clouds lead to substantial biases of GCM radiative quantities (Cahalan et al., 1994a, 1994b; Cairns et al., 2000). Further, an assumption of how partial cloudiness vertically overlaps within each GCM grid column is required. The widely employed assumption is the maximum-random overlap (Geleyn and Hollingsworth, 1979), advocated by many studies (e.g., Morcrette and Fouquart, 1986; Tian and Curry, 1989) and recently criticized by others, since it breaks down in case of vertically developed cloud systems

in strongly sheared environments (e.g., Hogan and Illingworth, 2000; Naud et al., 2008; Di Giuseppe and Tompkins, 2015). Last but not least, three-dimensional (3-D) radiative effects related to sub-grid horizontal photon transport, which in reality manifests itself most pronouncedly in regions characterized by strong horizontal gradients of optical properties, such as cloud side boundaries (Jakub and Mayer, 2015, 2016; Klinger and Mayer, 2014, 2016), are currently still neglected in the majority of GCMs. This broad palette of issues is challenging to tackle and solve.

In order to reduce the most striking plane-parallel biases, several methods were developed in the past. The scaling factor method, proposed by Cahalan et al. (1994a) and implemented in the ECMWF model by Tiedke (1996), was a conventional approach, where the cloud optical depth was multiplied by a constant factor and the resulting effective optical depth was passed to the radiation scheme. Barker (1996) introduced a more sophisticated gamma-weighted radiative transfer scheme, later also applied by Carlin et al. (2002) and Rossow et al. (2002), where the optical depth across a grid box is weighted using a

gamma distribution. Moreoever, Barker et al. (2002) and subsequently Pincus et al. (2003) presented an alternative technique, known as the Monte Carlo integration of Independent Column Approximation (McICA; Fig. 1, bottom middle), which is currently operationally employed in most large-scale atmospheric models. The fundamental assumption of the McICA is that the Independent Column Approximation (ICA; Fig. 1, top right) is adequate and therefore allows for the independent generation of sub-grid cloudy columns, which is managed by means of stochastic cloud generator (Räisänen et al., 2004; Räisänen and

Barker, 2004). As the full ICA is not affordable within the computational constraints of simulating complex weather and climate scenarios, the computing speed gain in the McICA approach is based on the simultaneous sampling of sub-grid cloud state and spectral interval.

Whereas all aforementioned methodologies certainly brought improvements compared to the conventional plane-parallel cloud representation, they all have some disadvantages. The usage of the McICA, for example, introduces conditional random

errors (the McICA noise) to radiative quantities and it is unclear, how significantly this affects the forecast skill. Räisänen et al. (2007), as an illustration, investigated the impact of the McICA noise in an atmospheric GCM (ECHAM5, Roeckner et al., 2003) and found statistically discernible impacts on simulated climate for a fairly reasonable McICA implementation. The largest effect was observed in the boundary layer, where clouds are essentially maintained by local cloud top radiative cooling. As the McICA noise disrupted this cooling, a positive feedback loop was induced, where a reduction of cloud fraction lead to

weaker radiative cooling, which in turn further diminished the cloud fraction. Similar findings were already previously reported by Räisänen et al. (2005) for global climate simulated with another GCM.

A few years after the introduction of the McICA, Shonk and Hogan (2008) [hereafter abbreviated to SH08] proposed a unique method, which utilizes two regions in each vertical model layer to represent the cloud, as opposed to one. One region is used





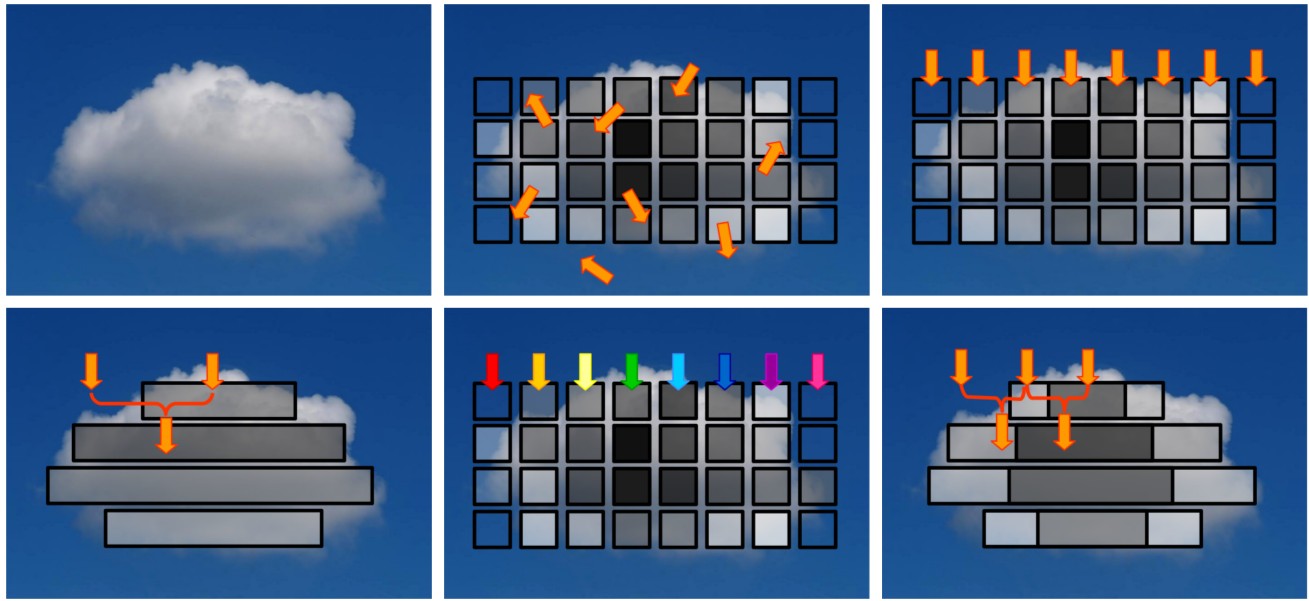

**Figure 1.** Divergent modeling of cloud-radiation interaction (arrows denote radiative fluxes; grey shading mirrors cloud optical thickness): top middle − realistic 3-D radiation calculation on a high-resolution cloud; top right − the ICA approximation; bottom left − the conventional plane-parallel approach in coarse-resolution weather and climate models; bottom middle − the McICA algorithm (rainbow colored fluxes indicate calculations in various spectral bands); bottom right − the Tripleclouds methodology.

to represent the optically thicker part of the cloud and the other represents the remaining optically thinner part − the method therefore captures cloud horizontal inhomogeneity. Together with the cloud-free region, the radiation scheme thus has three regions at each height and is referred to as the "Tripleclouds" (TC). In the pioneer work of SH08 the layer cloudiness was split into two equally-sized regions and the corresponding pair of cloud condensates (e.g., liquid water content, LWC) was generated on the basis of known LWC distribution. The method was initially tested on high-resolution radar data, where the exact position of the three regions was passed to the radiative solver, capable of representing an arbitrary vertical overlap. In practice, a host GCM usually provides only mean LWC and no information about vertical cloud arrangement. In order to make the method applicable to GCMs, Shonk et al. (2010) derived a global estimate of cloud horizontal variability in terms of fractional standard deviation (FSD), which can be used to split the mean LWC into two components along with the LWC distribution assumption. Further, they incorporated a generalized vertical overlap parameterization, called the exponential-random overlap, accounting for the aforementioned problematics in strongly sheared conditions. Recently, the method was successfully implemented in the *ecRad* package (Hogan and Bozzo, 2018), the current radiation scheme of ECMWF Integrated Forecast System (IFS). In contrast to the McICA, which is still operational also at ECMWF, the TC scheme does not produce any radiative noise. As suggested by Hogan and Bozzo (2016) this superiority could become even more valuable in the future if an alternative gas-



optics model with fewer spectral intervals than the current RRTM-G (Mlawer et al., 1997) will be developed, since this would increase the level of the McICA noise, but it would not affect the Triipleclouds.

Before the TC solver can be operationally employed, however, it has to be further validated. Whereas all previous studies employing the TC scheme examined primarily the top-of-the-atmosphere (TOA) radiation budget, the present work is aimed at evaluating the atmospheric heating rate and net surface flux. To that end, building upon the Triipleclouds idea, we developed the $\delta$-Eddington two-stream method for two cloudy and one cloud-free region at each height (Fig. 1, bottom right). The prime focus of this paper is to document our exertion of the Triipleclouds concept into the two-stream framework as well as the subsequent implementation of the radiative solver in the comprehensive radiative transfer package *libRadtran* (Mayer and Kylling, 2005; Emde et al., 2016). Another aim of the present study is to explore the TC potential for shallow cumulus clouds, applying various solver configurations diagnosing atmospheric heating rate and net surface flux. The challenge is to optimally set the condensate pair characterizing the two cloudy regions and geometrically split the layer cloudiness. We test the validity of global FSD estimate in conjunction with three assumptions for sub-grid cloud condensate distribution, which is of practical importance for the application in weather and climate models.

The manuscript is organized as follows: In Sect. 2 we introduce the shallow cumulus case study as well as preliminary radiative transfer experiments, demonstrating the importance of representing cloud horizontal heterogeneity. In Sect. 3 we present our version of the TC radiation scheme. Sect. 4 revises existing methodologies for generating cloud condensate pair. The TC performance is evaluated in Sect. 5. A brief summary and concluding remarks are given in Sect. 6.

## 2 Cloud data and methodology

### 2.1 Shallow cumulus clouds

#### 2.1.1 Core-shell model for convective clouds

We provide a brief note regarding the horizontal distribution of cloud condensate in convective cloud systems. This knowledge will be exploited later when constructing the Triipleclouds radiation scheme. Shallow cumulus clouds are convective clouds, which are often treated with the "core-shell model" (Heus and Jonker, 2008; Heiblum et al., 2019). In this model the convective "cloud core" associated with updraft motion and increased condensate loading is located in the geometrical centre of the cloud, surrounded by the "cloud shell" associated with downdrafts and condensate evaporation. The core-shell model is supported by multiple observational studies (e.g., Heus et al., 2009; Rodts et al., 2003; Wang et al., 2009) and numerical modelling investigations (e.g., Heus and Jonker, 2008; Jonker et al., 2008; Seigel, 2014) and hence represents the essence of several convection parametrizations. Heiblum et al. (2019) showed that the core-shell model is valid for about 90 % of the typical cloud's lifetime, with the largest discrepancy from the assumed core-shell geometry occurring during the dissipation stage of the cloud. Whereas most of the clouds contain a single core, larger clouds can possess multiple cores. Similarly, clouds in a cloud field have multiple cores, whereby their aggregate effect can be modelled with a core-shell model (Heiblum et al., 2019).





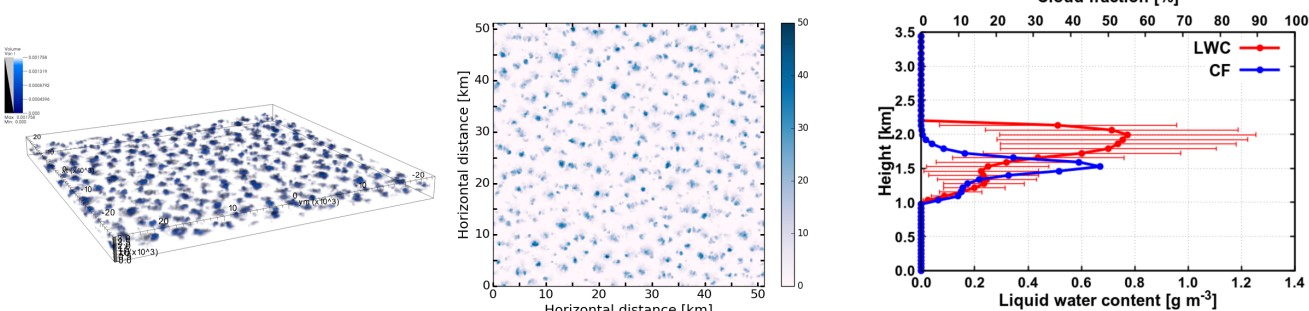

**Figure 2.** Left − shallow cumulus cloud field used as input for radiative transfer calculations (visualization with VisIt; Childs et al., 2012). Middle − vertically integrated optical thickness in the visible spectral range. Right − averaged LWC, its standard deviation (marked with errorbars) and cloud fraction.

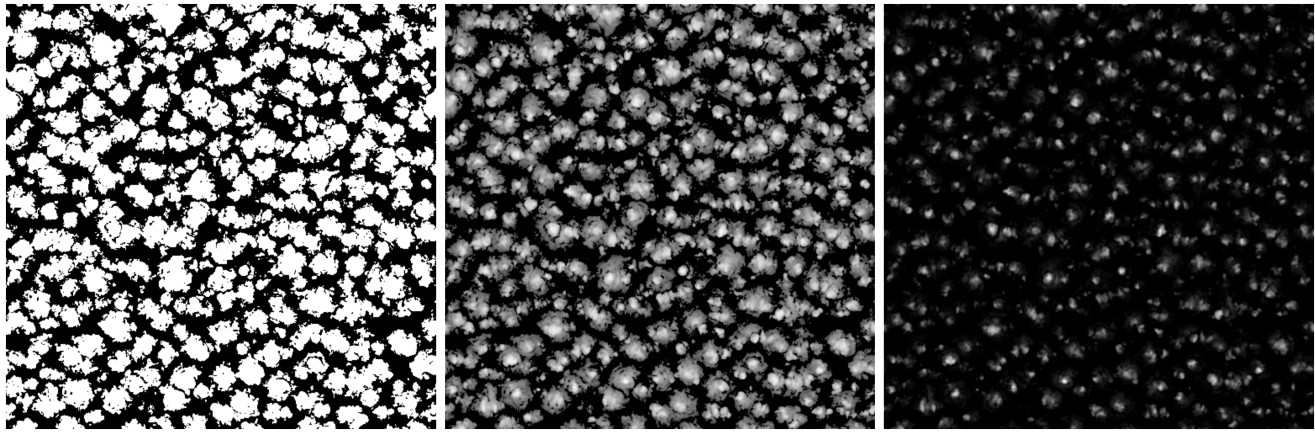

**Figure 3.** Horizontal heterogeneity for shallow cumulus cloud field. Left − cloud mask (clouds in white, clear-sky in black). Middle and right − vertically integrated optical thickness (with increasing thresholds). The comparison of the three panels demonstrate that optically thicker convective cores are located in the interior of individual clouds.

### 2.1.2  Shallow cumulus cloud field case study

Input for radiative transfer calculations is a shallow cumulus cloud field with a total cloud cover of 54.8 % (visualized in Fig. 2), simulated with the University of California, Los Angeles large-eddy simulation (UCLA-LES) model (Stevens et al., 2005; Stevens, 2007). The horizontal domain size is 51.2 x 51.2 km$^2$, with the vertical extent of the domain being 3.5 km. A constant horizontal grid spacing of 100 m is applied, whereas the vertical grid spacing is variable ranging from 50 m at the ground to 84 m at domain top. Further details about the UCLA-LES setup can be found in Jakub and Mayer (2017). A 3-D LWC distribution

was extracted from a simulation snapshot and the corresponding effective radius ($R_e$) was parameterized according to Bugliaro et al. (2011). Figure 2 (middle) shows vertically integrated cloud optical thickness, demonstrating that optically thicker regions





are located in the interior of individual clouds, which conforms to the core-shell model (see also Fig. 3). Vertical profiles of averaged LWC, its standard deviation ($\sigma_{LWC}$; simplest measure of cloud horizontal inhomogeneity) and cloud fraction are shown on Fig. 2 (right).

## 2.2 Radiative transfer models and experimental setup

### 2.2.1 Radiative transfer models

The radiative transfer experiments were performed using the *libRadtran* software (www.libradtran.org), which contains several radiation solvers. The benchmark calculations were performed with the 3-D model MYSTIC, the Monte Carlo code for the physically correct tracing of photons in cloudy atmospheres (Mayer, 2009), which can be run in ICA mode as well. Further, we employed the classic $\delta$-Eddington two-stream method (Joseph et al., 1976) suitable for horizontally homogeneous layers (either fully cloudy or fully clear-sky) and the extension of this method, which allows for partial cloudiness. The latter is the $\delta$-Eddington two-stream method with maximum-random overlap assumption, which was recently implemented into *libRadtran* in the configuration as described in Črnivec and Mayer (2019) and is ideally siuted as a proxy for the conventional GCM radiative solver (additional instructions regarding its usage are provided in Appendix A).

### 2.2.2 Setup of radiative transfer experiments

The background thermodynamic state was the US standard atmosphere (Anderson et al., 1986). The parameterization of Hu and Stamnes (1993) was used to convert LWC and $R_e$ into cloud optical properties. The solar experiments were performed for a solar zenith angle (SZA) of $0°$, $30°$ and $60°$ and a surface albedo of 0.25. In the thermal part of the spectrum the surface was assumed to be nonreflective. The shortwave calculations applied 32 spectral bands of the correlated k-distribution by Kato et al. (1999), whereas the longwave calculations employed 12 spectral bands adopted from Fu and Liou (1992). In the Monte Carlo experiments the standard forward and the efficient backward photon tracing were employed in the solar and thermal spectral range respectively. The resulting Monte Carlo noise of domain-averaged quantities is negligible (less than 0.1 %).

### 2.2.3 Diagnostics and error calculation

The radiative diagnostics include atmospheric heating rate and net (difference between downward and upward) surface flux. Each diagnostic was examined in the solar, thermal (nighttime effect) and total (daytime effect) spectral range. The error is given by the absolute bias (Eq. 1), relative bias (Eq. 2) and for the atmospheric heating rate additionally by the root mean square error evaluated throughout the vertical extent of the cloud layer (Eq. 3):

$$\text{absolute bias} = y - x, \tag{1}$$

$$\text{relative bias} = \left(\frac{y}{x} - 1\right) \cdot 100\%, \tag{2}$$





$$\text{cloud-layer RMSE} = \sqrt{\overline{(y-x)^2}}, \tag{3}$$

where $y$ represents the biased quantity and $x$ represents the benchmark.

### 2.3 Preliminary radiative transfer experiments

We present a set of preliminary radiative transfer experiments (listed in Table 1), introducing the 3-D benchmark, the ICA and the conventional GCM calculation. Further, we aim to quantify the various error sources of GCM radiative heating rates, in particular the error related to neglected cloud horizontal heterogeneity.

#### 2.3.1 Benchmark heating rate

The benchmark calculation using MYSTIC (abbreviated to "3-D" experiment) was performed on the highly-resolved LES
cloud field (Fig. 4, left). Supposing that the entire LES domain is contained within one GCM column, the quantity of interest is a single vertical profile of radiative heating rate, thus results were horizontally averaged across the domain. Figure 5 (left) shows the resulting benchmark profiles.

In the solar experiment for overhead Sun (Fig. 5, top left) there is a large absorption of radiation in the cloud layer, resulting in a peak heating rate of 10.8 K day$^{-1}$. The latter is reached at a height of 1.6 km, which is slightly above the height of maximal
cloud fraction (Fig. 2, right). With decreasing Sun elevation the solar heating rate diminishes, exhibiting the maximum of 9.4 K day$^{-1}$ and 5.5 K day$^{-1}$ at SZA of 30° and 60°, respectively. The height where the peak heating is reached stays the same at all SZAs. In the thermal spectral range (Fig. 5, bottom left) the cloud layer is subjected to strong cooling, reaching a peak value of 17.7 K day$^{-1}$ attained at the same height as the maximum solar heating. Below this height, the magnitude of cooling decreases towards the cloud base, where a slight warming effect is observed.

#### 2.3.2 Conventional GCM representation

In order to mimic the conditions in conventional GCM models (Fig. 4, right), the cloud optical properties in each vertical layer were horizontally averaged over the cloudy part of the domain, creating a suite of plane-parallel partially cloudy layers. Consequently, the $\delta$-Eddington two-stream method with maximum-random overlap assumption was employed (abbreviated to "GCM" experiment).
The main shortcomings of the GCM compared to the benchmark (Fig. 5, right) are as follows. In the solar spectral range the peak heating rate is overestimated by 2.7, 2.1 and 0.8 K day$^{-1}$ at SZA of 0°, 30° and 60°, respectively. In the thermal spectral range the GCM bias artificially enhances radiatively driven destabilization of the cloud layer by an overestimation of cooling by 6.0 K day$^{-1}$ at cloud-layer top and an overestimation of warming by 3.4 K day$^{-1}$ at cloud-layer bottom. The GCM error sources are multiple: the misrepresentation of realistic cloud structure, the neglected sub-grid horizontal photon transport as
well as the intrinsic difference between the Monte Carlo and two-stream radiative solvers.



**Table 1.** List of preliminary radiative transfer experiments and their abbreviations.

| Experiment | Abbreviation |
|---|---|
| 3-D Monte Carlo radiative model on LES cloud field | 3-D |
| ICA Monte Carlo radiative model on LES cloud field | ICA |
| $\delta$-Eddington two-stream method on LES cloud field | TSM |
| $\delta$-Eddington two-stream method on homogenized LES cloud field | HOM |
| $\delta$-Eddington two-stream method with maximum-random overlap | GCM |

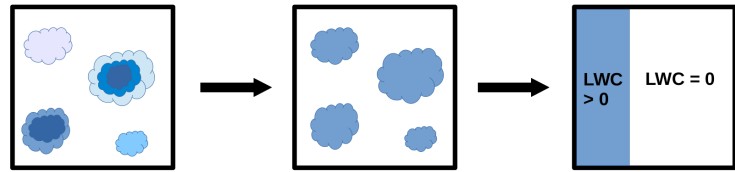

**Figure 4.** Left − a horizontal cross-section of LES cloud field. Middle − derived "homogenized" cloud field, which retains its 3-D geometry, but where horizontal heterogeneity is completely removed by applying averaged cloud optical properties in each vertical layer. Right − conditions in a grid box of a conventional GCM (homogeneous fractional cloudiness).

### 2.3.3 ICA and its limitations

To quantify the effect of neglected horizontal photon transport, we run the Monte Carlo radiative model in independent column mode on the original cloud field preserving its LES resolution (Fig. 4, left), with the result horizontally averaged over the domain (abbreviated to "ICA" experiment). Similarly, we applied the $\delta$-Eddington two-stream method within each independent column of the original LES grid (Fig. 4, left) and subsequently averaged the result horizontally (abbreviated to "TSM" experiment). The difference between the ICA and 3-D is a measure of horizontal photon transport. The difference between the TSM and 3-D is a measure of both the horizontal photon transport as well as the intrinsic difference between the Monte Carlo and two-stream radiative solvers.

As anticipated, both indepedent column experiments (ICA, TSM) perform similarly (Fig. 5, right), implying that the intrinsic difference between the radiative solvers is small. Therefore only the ICA is discussed hereafter. The solar bias increases with descending Sun (cloud side illumination; Hogan and Shonk, 2013; Jakub and Mayer, 2015, 2016), reaching a maximum of $-0.7$ K day$^{-1}$ at SZA of $60°$. The amount of thermal cooling is underestimated in the ICA (up to 1 K day$^{-1}$), since realistic cloud side cooling is neglected (Klinger and Mayer, 2014, 2016). Nevertheless, the ICA still overall performs considerably better than the conventional GCM, implying that the major error source of GCM heating rate stems from the misrepresentation of cloud structure, and not from the neglected horizontal photon transport.

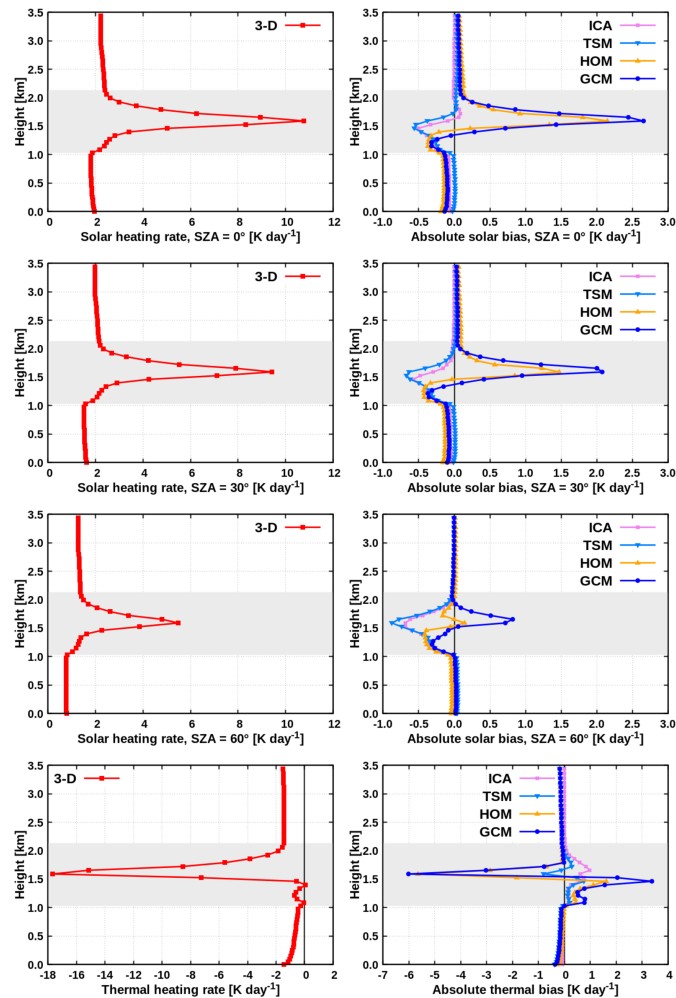

**Figure 5.** Radiative heating rate in preliminary experiments. The cloud layer is shaded grey.

### 2.3.4 Cloud horizontal heterogeneity effect

In order to isolate the effects of neglected cloud horizontal heterogeneity in a conventional GCM from other effects related to the misrepresentation of cloud structure (e.g., vertical overlap assumption), we employed the GCM radiative solver on the cloud field preserving its LES resolution, but with removed horizontal heterogeneity (Fig. 4, middle). In this way the averaged (plane-parallel) cloud optical properties were applied in each vertical layer, but the realistic 3-D cloud field geometry was retained. The results were horizontally averaged (abbreviated to "HOM" experiment).

The radiative heating rate in the HOM experiment (Fig. 5, right) is to a great extent similar to that in the GCM (especially in the solar experiments at SZA of 0° and 30° as well as in the thermal experiment), implying that the dominant GCM error source is indeed the neglected cloud horizontal heterogeneity. The question that we attempt to answer is: how much of this bias





can be removed with Tripleclouds? In other words, how well can the continuous probability density function (PDF) of layer LWC be represented by just two cloudy regions (a two-point PDF)?

## 3 The Tripleclouds radiative solver

We first explain in Sect. 3.1 the $\delta$-Eddington two-stream radiation scheme and introduce the terminology. We focus only on those aspects of the method, important to understand its extension to multiple (three) regions, explained in subsequent Sect.

3.2 and 3.3. Differences between the radiative solver of SH08 and our implementation are summarized in Sect. 3.4. Further technical instructions regarding the Tripleclouds usage within the scope of *libRadtran* are provided in Appendix A.

### 3.1 $\delta$-Eddington two-stream method

In the classic two-stream approach, the entire radiative field is approximated solely with direct solar beam ($S$) and two streams of diffuse radiation: the downward ($E_\downarrow$) and upward ($E_\uparrow$) component. The widely employed $\delta$-Eddington approximation is a

reliable way to account for a strong forward-scattering peak of cloud droplets (Joseph et al., 1976; King and Harshvardhan, 1986; Stephens et al., 2001). For the calculations in a vertically inhomogeneous atmosphere, the atmosphere is divided into a number of homogeneous layers, each characterized by its set of constant optical properties. Considering a single layer ($j$) located between levels ($i$-1) and ($i$) (illustrated in Fig. 6)[1], a system of linear equations determining the fluxes emanating from the layer as a function of fluxes entering the layer can be written as:

$$
\begin{pmatrix} E_\uparrow(i-1) \\ E_\downarrow(i) \\ S(i) \end{pmatrix} = \begin{pmatrix} a_{11} & a_{12} & a_{13} \\ a_{12} & a_{11} & a_{23} \\ 0 & 0 & a_{33} \end{pmatrix} \cdot \begin{pmatrix} E_\uparrow(i) \\ E_\downarrow(i-1) \\ S(i-1) \end{pmatrix}. \quad (4)
$$

The coefficients $a_{kl}$ in Eq. (4) are referred to as Eddington coefficients. They depend on the optical properties of layer ($j$) and have the following physical meaning:

– $a_{11}$ - transmission coefficient for diffuse radiation,

– $a_{12}$ - reflection coefficient for diffuse radiation,

– $a_{13}$ - reflection coefficient for the primary scattered solar radiation,

– $a_{23}$ - transmission coefficient for the primary scattered solar radiation,

– $a_{33}$ - transmission coefficient for the direct solar radiation.

---

[1]We follow the convention of $i, j$ increasing downward from the top of the atmosphere, where $i = 0$, $j = 1$. Index $i$ is used for level variables, while index $j$ is used for layer variables. The $N$ vertical layers, enumerated from 1 to $N$, are enclosed by ($N$+1) vertical levels, enumerated from 0 to $N$.





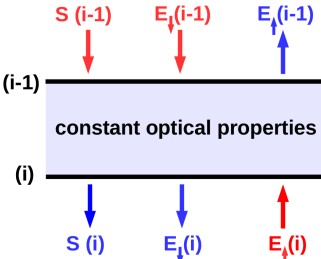

**Figure 6.** A homogeneous model layer between levels ($i$-1) and ($i$). Incoming radiative fluxes are coloured red, outgoing fluxes are colored blue.

For the inclusion of thermal radiation in Eq. (4), the reader is referred to Zdunkowski et al. (2007). The individual layers are coupled vertically by imposing flux continuity at each level. Taking the boundary conditions at TOA (Eq. 5) and at the ground

(Eq. 6, with $A_g$ representing ground albedo) into account,

$$E_\downarrow(0) = 0, \tag{5}$$

$$E_\uparrow(N) = A_g[S(N) + E_\downarrow(N)], \tag{6}$$

the radiative fluxes throughout the atmosphere are computed by solving the matrix problem (Coakley and Chylek, 1975;

Wiscombe and Grams, 1976; Meador and Weaver, 1980; Ritter and Geleyn, 1992). Henceforth, the calculation of heating rates is straightforward.

### 3.2 $\delta$-Eddington two-stream method for three regions at each height

Consider now a model layer located between levels ($i$-1) and ($i$) divided into three regions (Fig. 7). Such layer is characterized by three sets of optical properties and corresponding Eddington coefficients: one for the region of optically thick cloud (super-

script "$ck$"), the other for the region of optically thin cloud (superscript "$cn$") and the third for the cloud-free region (superscript "$f$"). In order to apply vertical overlap rules the radiative fluxes corresponding to each of the three regions need to be defined separately at each level (e.g., $S^{ck}$, $S^{cn}$ and $S^f$; and analogously for both diffuse components). Total radiative flux at level ($i$) is thus the sum of both cloudy and the cloud-free component:

$$S(i) = S^{ck}(i) + S^{cn}(i) + S^f(i), \tag{7}$$


$$E_\downarrow(i) = E_\downarrow^{ck}(i) + E_\downarrow^{cn}(i) + E_\downarrow^f(i), \tag{8}$$

$$E_\uparrow(i) = E_\uparrow^{ck}(i) + E_\uparrow^{cn}(i) + E_\uparrow^f(i). \tag{9}$$





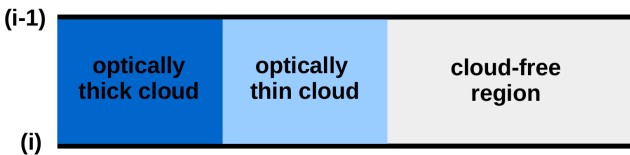

**Figure 7.** A model layer between levels $(i\text{-}1)$ and $(i)$ divided into three regions.

The equation system (4) is replaced by:

$$
\begin{pmatrix} E_\uparrow^{ck}(i-1) \\ E_\downarrow^{ck}(i) \\ S^{ck}(i) \end{pmatrix} = \begin{pmatrix} a_{11}^{ck} & a_{12}^{ck} & a_{13}^{ck} \\ a_{12}^{ck} & a_{11}^{ck} & a_{23}^{ck} \\ 0 & 0 & a_{33}^{ck} \end{pmatrix} \cdot \begin{pmatrix} T_\uparrow^{ck,ck} E_\uparrow^{ck}(i) + T_\uparrow^{cn,ck} E_\uparrow^{cn}(i) + T_\uparrow^{f,ck} E_\uparrow^{f}(i) \\ T_\downarrow^{ck,ck} E_\downarrow^{ck}(i-1) + T_\downarrow^{cn,ck} E_\downarrow^{cn}(i-1) + T_\downarrow^{f,ck} E_\downarrow^{f}(i-1) \\ T_\downarrow^{ck,ck} S^{ck}(i-1) + T_\downarrow^{cn,ck} S^{cn}(i-1) + T_\downarrow^{f,ck} S^{f}(i-1) \end{pmatrix}, \quad (10)
$$

$$
\begin{pmatrix} E_\uparrow^{cn}(i-1) \\ E_\downarrow^{cn}(i) \\ S^{cn}(i) \end{pmatrix} = \begin{pmatrix} a_{11}^{cn} & a_{12}^{cn} & a_{13}^{cn} \\ a_{12}^{cn} & a_{11}^{cn} & a_{23}^{cn} \\ 0 & 0 & a_{33}^{cn} \end{pmatrix} \cdot \begin{pmatrix} T_\uparrow^{ck,cn} E_\uparrow^{ck}(i) + T_\uparrow^{cn,cn} E_\uparrow^{cn}(i) + T_\uparrow^{f,cn} E_\uparrow^{f}(i) \\ T_\downarrow^{ck,cn} E_\downarrow^{ck}(i-1) + T_\downarrow^{cn,cn} E_\downarrow^{cn}(i-1) + T_\downarrow^{f,cn} E_\downarrow^{f}(i-1) \\ T_\downarrow^{ck,cn} S^{ck}(i-1) + T_\downarrow^{cn,cn} S^{cn}(i-1) + T_\downarrow^{f,cn} S^{f}(i-1) \end{pmatrix}, \quad (11)
$$

$$
\begin{pmatrix} E_\uparrow^{f}(i-1) \\ E_\downarrow^{f}(i) \\ S^{f}(i) \end{pmatrix} = \begin{pmatrix} a_{11}^{f} & a_{12}^{f} & a_{13}^{f} \\ a_{12}^{f} & a_{11}^{f} & a_{23}^{f} \\ 0 & 0 & a_{33}^{f} \end{pmatrix} \cdot \begin{pmatrix} T_\uparrow^{ck,f} E_\uparrow^{ck}(i) + T_\uparrow^{cn,f} E_\uparrow^{cn}(i) + T_\uparrow^{f,f} E_\uparrow^{f}(i) \\ T_\downarrow^{ck,f} E_\downarrow^{ck}(i-1) + T_\downarrow^{cn,f} E_\downarrow^{cn}(i-1) + T_\downarrow^{f,f} E_\downarrow^{f}(i-1) \\ T_\downarrow^{ck,f} S^{ck}(i-1) + T_\downarrow^{cn,f} S^{cn}(i-1) + T_\downarrow^{f,f} S^{f}(i-1) \end{pmatrix}, \quad (12)
$$

so that the fluxes emanating from a certain region of the layer under consideration (e.g., region of optically thick cloud) generally depend on a linear combination of the incoming fluxes stemming from each of the three regions in adjacent layers. The coefficients starting with $T$ appearing in Eqs. (10), (11), (12) are referred to as the overlap (transfer) coefficients and correspond to layer $(j)$. The coefficient $T_\downarrow^{ck,cn}(j)$, for example, represents the fraction of downward radiation that leaves the base of optically thick cloud of layer $(j\text{-}1)$ and enters the optically thin cloud of layer under consideration $(j)$. The overlap coefficients quantitatively depend on the choice of the overlap rule, which will be discussed in the next subsection (3.3). For a three-region layer, the boundary condition at TOA (Eq. 5) implies:

$$
E_\downarrow^{ck}(0) = 0, \quad (13)
$$

$$
E_\downarrow^{cn}(0) = 0, \quad (14)
$$

$$
E_\downarrow^{f}(0) = 0. \quad (15)
$$





The boundary condition at the ground (Eq. 6) is extended to:

$$E_\uparrow^{ck}(N) = A_g[S^{ck}(N) + E_\downarrow^{ck}(N)], \tag{16}$$


$$E_\uparrow^{cn}(N) = A_g[S^{cn}(N) + E_\downarrow^{cn}(N)], \tag{17}$$

$$E_\uparrow^{f}(N) = A_g[S^{f}(N) + E_\downarrow^{f}(N)], \tag{18}$$

which assumes that the downward fluxes leaving the lowest model layer, after reflection enter the same sections of individual
cloudy and cloud-free air (isotropic ground reflection).

### 3.3   Overlap considerations

The layer cloud fraction $C$ is given by:

$$C(j) = C^{ck}(j) + C^{cn}(j). \tag{19}$$

In our implementation we demand the following relationship between the individual cloud fraction components:

$$C^{ck}(j) = \alpha \cdot C(j), \tag{20}$$

$$C^{cn}(j) = (1 - \alpha) \cdot C(j), \tag{21}$$

where $\alpha$ is a constant between 0 and 1. We apply the widely used maximum-random overlap assumption (Geleyn and
Hollingsworth, 1979) for the entire layer cloudiness (sum of optically thick and thin cloudy regions), where adjacent cloudy
layers exhibit maximal overlap and cloudy layers separated by at least one cloud-free layer exhibit random overlap. If the
cloudy layers are splitted into two parts, however, this overlap rule is not sufficient and needs to be extended. Therefore, we
additionally assume the maximum overlap of adjacent optically thicker cloudy regions and abbreviate this extended overlap
rule to the "maximum$^2$-random overlap". This assumption implicitly places the optically thicker cloudy region towards the
interior of the cloud in the horizontal plane, which is in line with the core-shell model.
Now we can quantitatively determine the overlap coefficients in Eqs. (10), (11) and (12) for the maximum$^2$-random overlap.
We consider the transmission of downward radiation through two adjacent layers with partial cloudiness. Four possible geome-
tries, illustrated in Fig. 8, need to be treated. For the situation depicted on the top left panel of Fig. 8, the transmission of direct
radiation can be formulated as follows. The optically thick cloud of layer $(j\text{-}1)$ transmits $S^{ck}$ $(i\text{-}1)$, the optically thin cloud
transmits $S^{cn}$ $(i\text{-}1)$ and the cloud-free region transmits $S^{f}$ $(i\text{-}1)$. These three components of the transmitted radiation must then





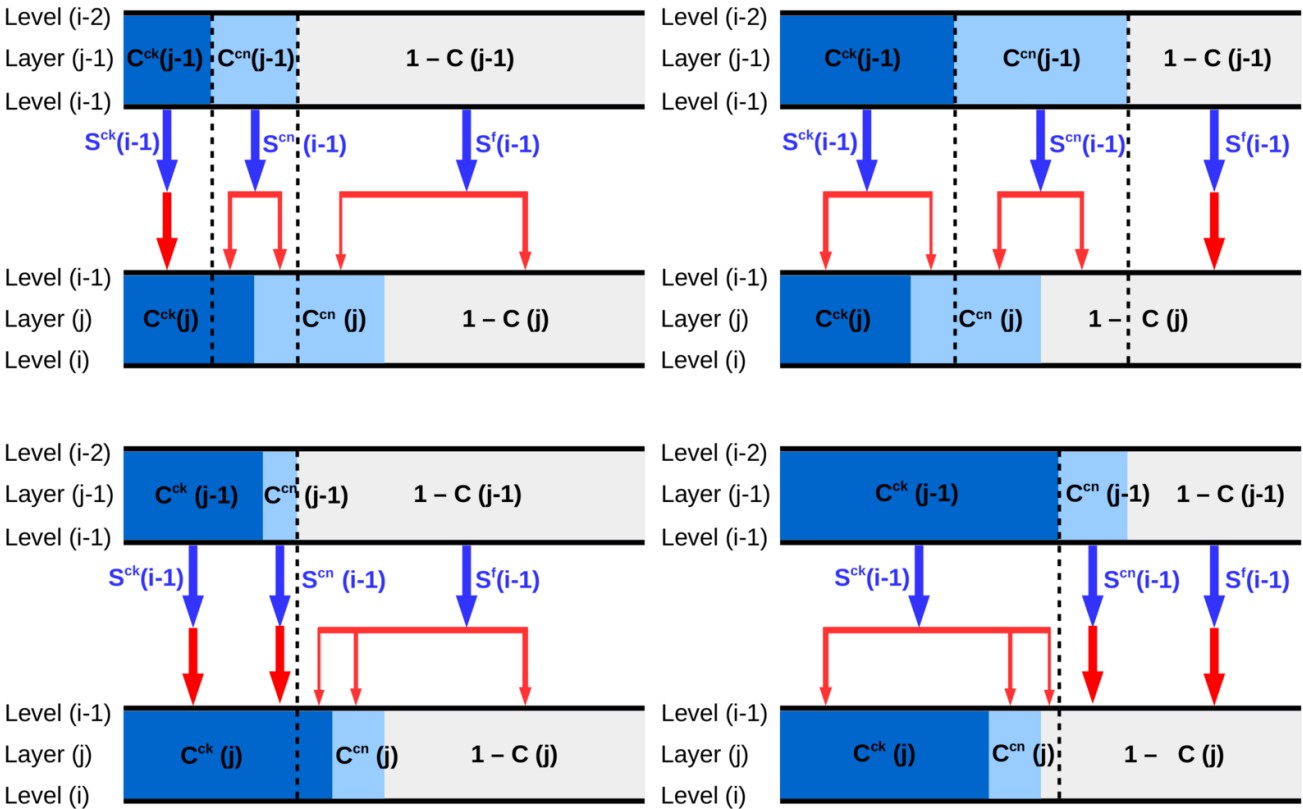

**Figure 8.** Transmission of direct solar radiation through two adjacent layers with partial cloudiness for the maximum[2]-random overlap concept.

be distributed between the three regions of the lower layer ($j$). The maximum overlap of optically thick cloudy regions implies that the entire radiation $S^{ck}$ leaving the base of layer ($j$-1) enters the optically thick cloud below:

$$T_{\downarrow}^{ck,ck}(j) = 1,\tag{22}$$

and none of it enters the other two regions:

$$T_{\downarrow}^{ck,cn}(j) = 0,\tag{23}$$

$$T_{\downarrow}^{ck,f}(j) = 0.\tag{24}$$

To ensure the maximum overlap of cloudy layers as a whole, the remaining cloudy flux at the base of layer ($j$-1), namely the $S^{cn}$ ($i$-1), needs to be lead into the two cloudy regions of the lower layer, with the priority to enter the optically thick cloud.





This yields:

$$T_\downarrow^{cn,ck}(j) = \frac{C^{ck}(j) - C^{ck}(j-1)}{C^{cn}(j-1)}, \tag{25}$$

$$T_\downarrow^{cn,cn}(j) = \frac{[C^{ck}(j-1) - C^{cn}(j-1)] - C^{ck}(j)}{C^{cn}(j-1)}, \tag{26}$$

$$T_\downarrow^{cn,f}(j) = 0. \tag{27}$$

The cloud-free flux $S^f$ at the base of layer ($j$-1) is distributed according to:

$$T_\downarrow^{f,f}(j) = \frac{1 - C(j)}{1 - C(j-1)}, \tag{28}$$

$$T_\downarrow^{f,cn}(j) = \frac{C(j) - C(j-1)}{1 - C(j-1)}, \tag{29}$$

$$T_\downarrow^{f,ck}(j) = 0. \tag{30}$$

We leave the derivation of overlap coefficients for other three geometries as an exercise for the reader. The transmission of upward radiation is managed via overlap coefficients $T_\uparrow^{a,b}$ in a similar fashion. It should be noted that the same coefficients govern the reflection, whereby the upward reflection of downward radiation is treated with $T_\downarrow^{a,b}$ and the reverse situation is treated with $T_\uparrow^{a,b}$. Pairwise overlap as employed here ensures that the matrix problem is fast to solve. Whereas a drawback of the core-shell model and thereby the outlined overlap is that it underperforms in case of vertically developed cloud systems in strongly sheared conditions, the present Tripleclouds implementation is an excellent tool to study shallow convective clouds. In this way the effects of cloud horizontal inhomogeneity are tackled in isolation, while the issues related to vertical shear are eliminated.

The Tripleclouds radiative solver has been successfully implemented in the *libRadtran* package. Technically, the calculation of overlap coefficients is performed in an autonomous function enabling flexible modifications of overlap rules in the future.

### 3.4 Differences to the radiative solver of SH08

We briefly outline the main differences between our radiative solver and that of SH08 regarding the incorporation of three-region layers in two-stream equations. SH08 used another version of a two-stream solver implemented in the radiation scheme devised by Edwards and Slingo (1996). The way, how the two-stream solver was incorporated in the Edwards-Slingo code, resulted in a complicated expression for upward fluxes, when the solver was extended to multiple regions (their Eq. 15). Therefore SH08 simplified this expression, achieving higher computational efficiency, but bringing certain physical shortcomings.





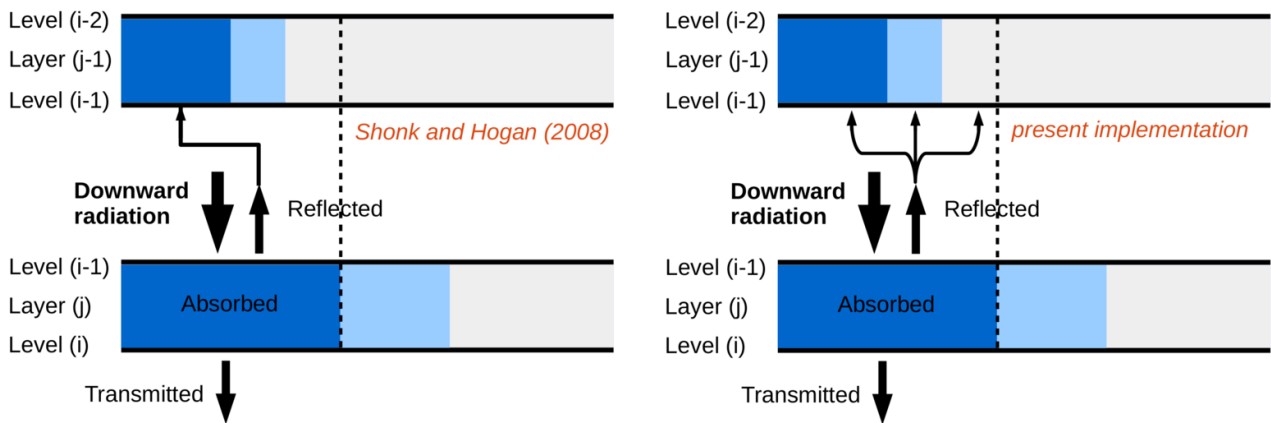

**Figure 9.** Illustration of the differences between the radiative solver introduced by SH08 (left) and our implementation (right) regarding the treatment of upward reflection of downward radiation. The schematic exemplifies the treatment of downward radiation entering the optically thick cloudy region of the layer (j) under consideration, although analogous considerations (overlap rules) are applied to the other two regions in the layer as well.

They explained: "Physically, this means that, at each height, the downwelling radiation in a particular region is either reflected back or absorbed; none is reflected up into another region."

The Tripleclouds radiative solver as constructed in the present work allows for a full interaction of the three regions, meaning

that downward radiation in a particular region being reflected upward (and vice versa), can in general be distributed between all three regions. This feature is illustrated in Fig. 9 and could play a comparatively important role especially in the solar calculations, because of significant scattering effects. Moreover, this physical mechanism was recently recognized as relevant and incorporated also into the latest version of the Tripleclouds solver at ECMWF (Hogan et al., 2019). Whereas the work of Hogan et al. (2019) represents the first study utilizing the complete TC form researching the TOA cloud radiative forcing, the

present study pioneeringly applies the full TC scheme to atmospheric heating rate and net surface flux.

## 4   Methodologies to generate the LWC pair

In order to apply the TC radiative solver, a pair of LWC characterizing optically thin and thick cloudy regions (LWC$^{cn}$, LWC$^{ck}$) needs to be created in each vertical layer. In Sect. 4.1 we revise the original Tripleclouds method introduced by SH08, later referred to as the "lower percentile method" (Shonk et al., 2010), which can only be applied if the LWC distribution is known.

In Sect. 4.2 we summarize the more practical "fractional standard deviation method" (Shonk et al., 2010).



## 4.1 The lower percentile method

In this method it is assumed that the LWC distribution in each vertical layer can be approximated with the normal distribution:

$$p(LWC) = \frac{1}{\sqrt{2\pi}\sigma_{LWC}} \exp\left[-\frac{(LWC - \overline{LWC})^2}{2\sigma_{LWC}^2}\right], \tag{31}$$

where $\overline{LWC}$ is layer mean LWC and $\sigma_{LWC}$ is its standard deviation. The distribution of LWC is divided into two regions through a given percentile of the distribution, denoted as "split percentile (SP)". The latter is chosen to be the 50th percentile or the median, which splits the cloud volume into two equal parts (i.e., cloud fraction in each vertical layer is halved). The LWC of the optically thin cloud ($LWC^{cn}$) is determined as the value corresponding to the so-called "lower percentile (LP)" of the distribution. This is chosen to be the 16th percentile based on the following considerations. We adjust the two LWC values in a way that the mean LWC in the layer is conserved:

$$\overline{LWC} = \frac{LWC^{ck} + LWC^{cn}}{2}, \tag{32}$$

and that they are separated by two standard deviations:

$$LWC^{ck} - LWC^{cn} = 2\sigma_{LWC}. \tag{33}$$

For a Gaussian distribution, the latter constraint has a desired property that the variability within each of the two cloudy regions (measured by $\sigma_{LWC}$) is the same as that within the entire cloud in the layer. Equations (32) and (33) give the following relationship for $LWC^{cn}$:

$$LWC^{cn} = \overline{LWC} - \sigma_{LWC}. \tag{34}$$

The fraction of the distribution with LWC lower than $LWC^{cn}$ is therefore:

$$f_{cn} = \int_{-\infty}^{LWC^{cn}} p(LWC)dLWC = 0.159, \tag{35}$$

which corresponds to the LP of 16. Finally, the $LWC^{ck}$ is determined using Eq. (32) to conserve the mean. Figure 10 shows the resulting LWC pair when the LP method is applied on shallow cumulus cloud field.

It should be noted that the choice of the 16th percentile as the LP and the 50th percentile as the SP is based solely on theoretical considerations. In practice, the LP and SP are the two tunable parameters, that can be adjusted according to their performance on real cloud data. Even though the optimal setting varies, SH08 exposed that the combination of LP of 16 and SP of 50 generally serves well in both solar and thermal spectral range for vast ranges of cloud data.

## 4.2 Fractional standard deviation method

This method in its initial formulation by Shonk et al. (2010) implicitly assumes that LWC is normally distributed as well. Thereby the cloudiness in each vertical layer is partitioned into two regions of equal size and the pair of LWC ($LWC^{cn}$,





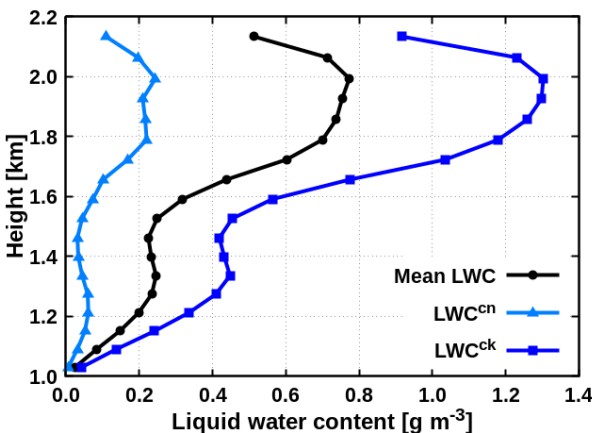

**Figure 10.** LWC profiles obtained with the LP method.

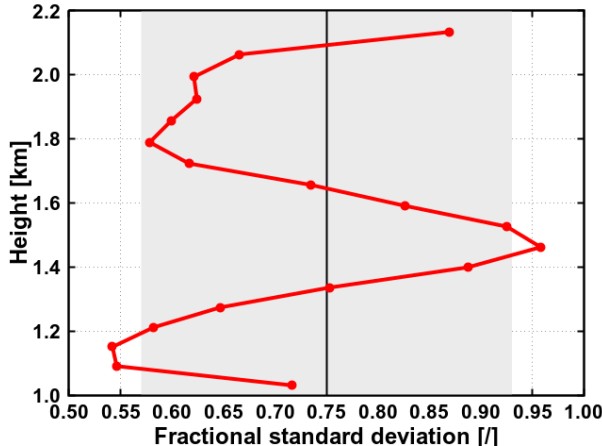

**Figure 11.** The actual FSD of the shallow cumulus. The grey-shaded area represents the uncertainty of global FSD estimate, centered around its mean value (black line).

LWC$^{ck}$) is obtained by:

$$LWC^{ck,cn} = \overline{LWC} \pm \sigma_{LWC} = \overline{LWC}(1 \pm FSD), \tag{36}$$

where FSD represents the fractional standard deviation of LWC:

$$FSD = \frac{\sigma_{LWC}}{\overline{LWC}}. \tag{37}$$

Since in practice only $\overline{LWC}$ is known within a GCM grid box, the FSD has to be parameterized. A review of numerous studies (Cahalan et al., 1994a; Barker et al., 1996; Pincus et al., 1999; Smith and DelGenio, 2001; Rossow et al., 2002; Hogan and Illingworth, 2003; Oreopoulos and Cahalan, 2005; SH08) carried out by Shonk et al. (2010) gave a globally representative





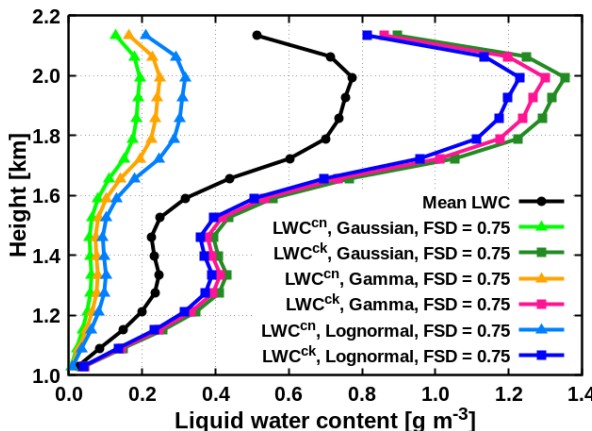

**Figure 12.** LWC profiles obtained with the FSD method using mean global estimate and altering LWC distribution.

FSD of $0.75 \pm 0.18$. Figure 11 shows the actual FSD for the present shallow cumulus: although this FSD is strongly dependent on the position within the cloud layer, it predominantly lies within the range of global estimate.

If the cloud condensate is normally distributed, substracting $\sigma_{LWC}$ from the $\overline{LWC}$ to obtain the $LWC^{cn}$ in Eq. (36) corresponds approximately with the 16th percentile. For more realistic lognormal and gamma distributions, the 16th percentile (advocated by SH08) is given by relationships presented in Hogan et al. (2016, 2019), whereby the $LWC^{ck}$ is again obtained by conserving the layer mean.

In order to test the validty of global FSD estimate, we applied its mean value (0.75) to create the pair of LWC in each vertical layer containing cloud. Further, to test the sensitivity of TC radiative quantities to the assumed form of the sub-grid cloud condensate distribution, we employed the FSD method in conjunction with all three distributions (the resulting profiles are shown in Fig. 12).

# 5   Application

We evaluated the TC radiative solver with both LP and FSD methods. The effective radii characterizing the two cloudy regions were kept the same (averaged $R_e$). The setup of radiation calculations was as described in Sect. 2.2. The results of the various TC experiments are compared with the conventional GCM, which approximates the cloud condensate distribution with a one-point PDF and can be perceived as an upper bound for the tolerable TC error. In addition, the ICA, which resolves the full sub-grid PDF, is shown as well. In Sect. 5.1 we examine atmospheric heating rate, whereas in Sect. 5.2 we discuss net surface flux.



### 5.1 Atmospheric heating rate

#### 5.1.1 Tripleclouds with LP method

We evaluate first the TC radiative solver when the LP method is used to obtain the pair of LWC. The results of this experiment,
denoted as "TC(LP)", are shown in Figs. 13 and 14. It is apparent that the TC(LP) is overall significantly more accurate than
the GCM. In the solar spectral range for overhead Sun (Fig. 13, top middle), the maximal bias within the cloud layer is reduced
from 2.7 K day$^{-1}$ to only 0.7 K day$^{-1}$. Whereas the largest bias reduction is observed within the cloud layer, the heating rate
above and below the cloud layer is considerably improved as well, explained as follows. The non-homogeneous clouds have
lower mean shortwave albedo and absorptivity than the corresponding plane-parallel cloudiness with the same mean optical
depth (Fig. 2 of Cairns et al., 2000). This implies that the non-homogeneous cloud in the TC configuration reflects less of the
incoming solar radiation upward (leading to a reduction of the positive GCM bias above the cloud layer) and simultaneously
absorbs less radiation (leading to a reduction of the positive GCM bias in the cloud layer), compared to the homogeneous cloud
in the GCM. Consequently, more radiation is transmitted through the cloud layer and absorbed in the region below the cloud
layer in the TC experiment compared to that in the GCM, which reduces the negative GCM bias in this region. At SZA of 30°
the behaviour is qualitatively similar, with the maximal bias of 2.1 K day$^{-1}$ within the cloud layer reduced by a factor of 5.
At SZA of 60°, the maximal bias of 0.8 K day$^{-1}$ within the cloud layer becomes of the opposite sign, but is still smaller in
magnitude ($-0.4$ K day$^{-1}$), when the TC(LP) is applied in place of the conventional GCM. In the layer above and especially
below the cloud layer, however, the bias is slightly increased. Noteworthy, at all three SZAs, the 3-D radiation feature at cloud
base (increased heating due to surface reflection of radiation) can not be properly accounted for using the TC solver.

In the thermal spectral range (Fig. 13, bottom middle), the degree of artificially enhanced destabilization of the cloud layer,
arising from the overestimation of cloud top cooling and cloud base warming in the GCM, is drastically reduced when the
TC(LP) is applied, interpreted as follows. The non-homogeneous clouds have lower mean longwave emissivity and absorptivity
than the corresponding homogeneous clouds with the same mean optical depth. Thus the non-homogeneous cloud top in the TC
experiment emits less radiation compared to the homogeneous cloud top in the GCM configuration, which reduces the negative
GCM bias at cloud top. Similarly, the non-homogeneous cloud base in the TC experiment absorbs less of the radiation stemming
from the warmer atmospheric layers underneath the cloud, compared to the homogeneous cloud base in the conventional GCM,
which reduces the positive GCM bias at cloud base. As anticipated, in the region above and below the cloud layer, the difference
between the TC and the GCM is only marginal.

#### 5.1.2 Tripleclouds with FSD method

We first examine the TC(FSD) experiment when the gaussianity of cloud condensate is assumed − this experiment is consid-
erably more accurate than the conventional GCM as well. As an illustration, the daytime cloud-layer RMSE of 1.7 K day$^{-1}$ is
reduced to 0.3 K day$^{-1}$ at SZA of 60° (Fig. 14). Furthermore, the TC(FSD) experiment is even slightly more accurate than the
TC(LP) especially in the thermal spectral range and in the solar spectral range at SZA of 30° and 60°, whereas at SZA of 0°
the situation is reversed (Fig. 13, middle column). The largest discrepancy between the two TC experiments is observed in the



**Figure 13.** Left − benchmark radiative heating rate. Middle and right − bias for the ICA, GCM and TC experiments.

central part of the cloud layer and is attributed to the fact that the actual layer LWC distribution of the present shallow cumulus deviates from the assumed Gaussian distribution as well as that the actual FSD deviates from the assumed global estimate.

In order to further support these findings, theoretical distributions (see also Appendix B) were fitted to the actual LWC distribution in each vertical cloudy layer (as illustrated in Fig. 15) and the Kolmogorov-Smirnov test (Conover, 1971; Wilks, 1995) was used to assess the goodness of fit. It was found that the actual LWC distribution is best approximated with the gamma distribution (best fit in 55 % of cloudy layers), followed by the lognormal distribution, whereas the Gaussian distribution always





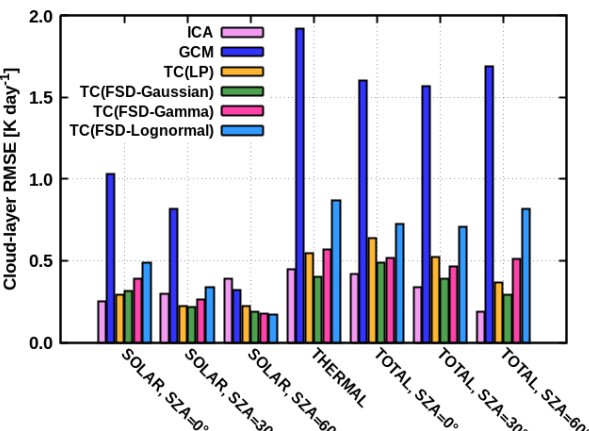

**Figure 14.** RMSE for the ICA, GCM and TC experiments.

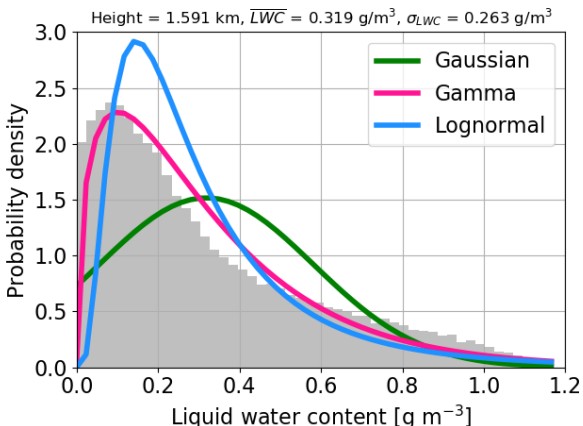

**Figure 15.** Actual LWC probability density in the central part of the cloud layer and distributional fits.

ranked worst. Precisely, the gamma distributional fit performed best throughout the central part of the cloud layer, where cloud-radiative effect is maximized.

When examining the entire set of TC(FSD) experiments it is apparent that the radiative heating rate is considerably more accurate compared to the conventional GCM regardless of the exact assumption for the LWC distribution. Although the Gaussian distribution was ranked worst when fitted to the actual PDF, the gaussianity assumption with global FSD performed best in practice, contemplated as follows. In the central part of the cloud layer around maximum cloud fraction the actual FSD of the present shallow cumulus (0.95) is larger than the assumed global estimate. The latter is primarily due to great amount of cloud side area in this region, an essential characteristic of broken cloud field, which generally contributes to increased variability (Hill et al., 2012, 2015). Since the assumption of gaussianity implies the largest difference between the LWC pair





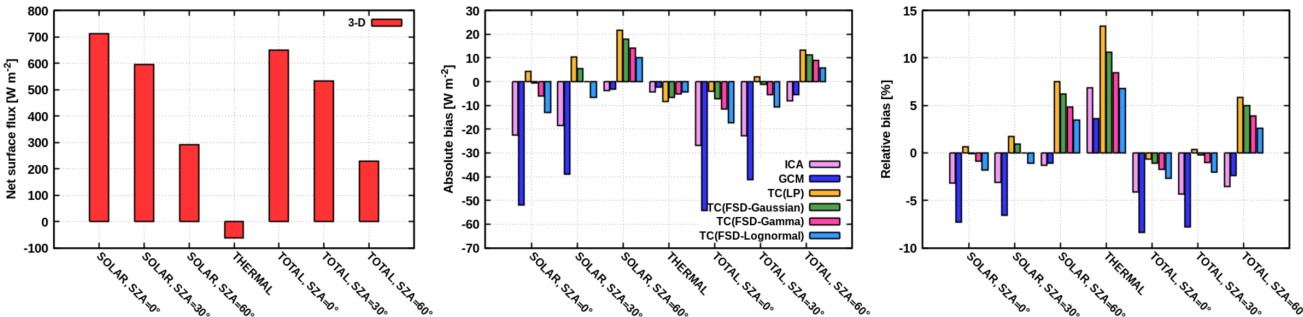

**Figure 16.** Left − benchmark net surface radiative flux. Middle and right − bias for the ICA, GCM and TC experiments.

characterizing the two cloudy regions (Fig. 12), it partially accounts for the missing variability provided by the global estimate.

More sophisticated FSD parameterizations are tempting to be tested in the future.

## 5.2 Net surface flux

Shallow cumulus clouds are a vital part of the planetary boundary layer, where the atmosphere is directly influenced by the presence of the Earth's surface. The net surface radiative flux is the key component of surface energy budget. The radiative biases at the surface, stemming from the inaccurate treatment of clouds, need to be properly understood and possibly best eliminated, as they generally feed back on the biases in the cloudy layers, when the radiation scheme is coupled to a dynamical model.

The behaviour of surface biases underneath the present shallow cumulus (Fig. 16, middle and right panel) is partially consistent with the findings gained when examining the cloud-layer heating rate error. In the ICA the daytime net surface flux is underestimated compared to 3-D at all SZAs. This is primarily due to well-acknowledged cloud side escape effect (Varnai and Davies, 1999; Hogan and Shonk, 2013), where the realistic scattering of radiation through cloud side areas increases 3-D downward surface radiation. Even when the Sun is lower in the sky (SZA of 60°) this mechanism overcomes the opposing cloud side illumination effect, where an elongated surface shadow reduces the 3-D net surface flux. Similarly, the strength of nocturnal surface cooling is overestimated in the ICA, since realistic cloud side emission is neglected.

The daytime GCM net flux bias at comparatively high Sun (SZA of 0° and 30°) is by a factor of 2 larger than the ICA bias. This is attributed to the fact that the plane-parallel GCM cloudiness leads to an increased solar absorption and hence reduced cloud-layer transmittance. The latter reduces downward flux reaching the surface and profoundly underestimates the net flux. At nighttime, the plane-parallel cloud in the GCM emits a greater amount of radiation towards the surface compared to heterogeneous cloud in the ICA, leading to a reduction of surface net flux bias.

When the Tripleclouds is applied either with the LP or the FSD method instead of conventional GCM radiation scheme, the daytime net surface flux bias of −55 W m$^{-2}$ (or −8 %) is substantially reduced to −5 W m$^{-2}$ (or −1 %) at overhead Sun and similarly for SZA of 30° (assuming gaussianity of cloud condensate). At SZA of 60° and especially at nighttime, radiative



bias in the various TC experiments increases compared to the GCM bias. This indicates that the TC in its current configuration should be taken with caution when applied to surface thermal flux, as its usage can lead to degradation of the nocturnal surface
budget compared to simple plane-parallel model.

## 6    Summary and conclusions

Inspired by the Tripleclouds concept of Shonk and Hogan (2008), we incorporated a second cloudy region in the widely used $\delta$-Eddington two-stream method with maximum-random overlap assumption for partial cloudiness. The resulting radiation scheme thus has two cloudy and one cloud-free region in each vertical layer and is capable of representing cloud horizontal
variability. The inclusion of a second cloudy region into the two-stream framework required an extension of vertical overlap rules. While retaining the maximum-random overlap for the entire layer cloudiness, we additionally assumed the maximum overlap of optically thicker cloudy regions in pairs of adjacent layers. This implicitly places the optically thicker region towards the interior of the cloud in the horizontal plane, while the optically thinner region resides at cloud periperhy, which is in line with the core-shell model for convective clouds.

The constructed Tripleclouds radiative solver was evaluated on a shallow cumulus cloud field. The validity of global estimate of fractional standard deviation − a common measure of cloud horizontal variability − was tested along with different assumptions for sub-grid cloud condensate distribution (Gaussian, gamma, lognormal), which are frequently applied when parameterizing clouds in weather and climate models. In the vast majority of experiments the Tripleclouds performed better than the conventional plane-parallel GCM scheme. The error of atmospheric heating rate was substantially reduced at day-
time and nighttime (up to fivefold cloud-layer RMSE reduction). In case of net surface flux the daytime bias was generally depleted as well, whereas the nighttime bias reduction was less pronounced, suggesting that the computationally more efficient plane-parallel scheme could be retained in this case.

    The question that needs to be addressed next is to what extent do our findings for a shallow cumulus case study with intermediate cloud cover apply to a larger set of scenarios comprising a wide range of cloud cover. This question is relevant,
because horizontal variability might essentially depend on cloud fraction. Similarly, the degree of cloud horizontal variability might depend on the GCM grid resolution, which has to be investigated in more detail in the future. Furthermore, organizational aspects of shallow convection should be addressed in the context of the present study. Mesoscale shallow convection sometimes occurs in the form of uniformly scattered cumuli, but is also frequently organized into cloud streets, clusters or mesoscale arcs (Agee et al., 1973; Atkinson and Zhang, 1996; Wood and Hartmann, 2006; Seifert and Heus, 2013). The robustness of the
results on the nature of cloud organization should be examined next. Recently, Stevens et al. (2019) proposed four mesoscale cloud patterns frequently observed in trade wind regions, which they labeled Sugar, Flower, Fish and Gravel. A follow-up study of Rasp et al. (2019) proved that the four patterns correspond to physically meaningful cloud regimes, each of them being associated with specific large-scale environmental conditions. These climatologically distinct environments should exhibit a highly variable cloud water variance. If this proves true and if the internal cloud variability is properly quantified, a regime-





dependent fractional standard deviation could be passed into Tripleclouds radiative solver in the next generation of global models.

An equivalent analysis then needs to be repeated for ice clouds. In order to carry out the analysis for clouds of large vertical growth, such as deep convective clouds, in a strongly sheared environment, the present vertical overlap rules have to be generalized. These topics are currently investigated by the corresponding author of this manuscript and will be discussed in detail in

upcoming studies.

*Code availability.*    The open-source UCLA-LES model is accessible at https://github.com/uclales. The *libRadtran* package is freely available at http://www.libradtran.org.

## Appendix A: Technical instructions for *libRadtran* users

The *libRadtran* radiative package is still under steady, continuous development. The latter goes hand in hand, inter alia, with

its plenty satisfied users worldwide. The core of the *libRadtran* package is the *uvspec* radiative transfer model, which contains several radiative transfer equation (RTE) solvers. To promote the usage of both recently implemented two-stream solvers (termed "`twomaxrnd`" and "`twomaxrnd3C`"), which are both coded in C programming language, basic guidelines are given below. For a complete description on how to set up the background atmosphere and other input parameters, the reader is referred to the *libRadtran* user manual, which is included in the software package. The output quantities of both algorithms

include either radiative fluxes (default) [W m$^{-2}$] or heating rates [K day$^{-1}$]. Whereas examples provided below illustrate the treatment of water clouds, both RTE solvers can be applied to ice clouds in a similar fashion.

## A1    RTE solver: "`twomaxrnd`"

The *δ-Eddington two-stream method with maximum-random overlap assumption for partial cloudiness* in the configuration as documented in Sect. 2.2 of Črnivec and Mayer (2019) is called as follows:


    **rte_solver** `twomaxrnd`
    **cloud_fraction_file** `cf.dat`
    **wc_file 1D** `wc.dat`

where `cf.dat` is the standard *libRadtran* file containing cloud fraction vertical profile and `wc.dat` is the standard 1-D file defining water cloud properties.





## A2   RTE solver: `"twomaxrnd3C"`

The *Tripleclouds radiative solver*, effectively the *δ-Eddington two-stream method for two cloudy and one cloud-free region at each height with maximum$^2$-random overlap assumption*, as described in Sect. 3 of the present work, is invoked as follows:


**rte_solver** `twomaxrnd3C`
**cloud_fraction_file** `cf.dat`
**twomaxrnd3C_scale_cf** `0.4`
**profile_file wck 1D** `wck.dat`
**profile_file wcn 1D** `wcn.dat`

where `cf.dat` is again the standard file containing the vertical profile of cloud fraction. It is important to note that this file determines the cloud fraction of the entire layer cloudiness (sum of optically thick and thin cloudy regions). The division of the latter into two components is managed via newly introduced parameter `twomaxrnd3C_scale_cf`, which corresponds to
the parameter $\alpha$ in Eqs. 20 and 21. The split of averaged cloud water properties into two components is not yet automated, rather the user is asked to preprocess both cloud files depending on his/her specific needs. The resulting `wck.dat` and `wcn.dat` are 1-D water cloud files, defining properties of optically thick and thin cloudy regions, respectively (note that the option `profile_file` is solely the generalization of the `wc_file` command).

## Appendix B: Analytical probability density functions

We outline the relationship between $\overline{LWC}$, $\sigma_{LWC}$, FSD (denoted as $f_{LWC}$ in the following) and the parameters used to describe lognormal and gamma distributions, which were applied to fit the modeled LWC distributions.

A lognormal distribution of LWC is defined as:

$$p(LWC) = \frac{1}{\sqrt{2\pi}\sigma_0 LWC} \exp\left[ -\frac{\ln(LWC/LWC_0)^2}{2\sigma_0^2} \right]. \tag{B1}$$

The parameters of the lognormal distribution, $LWC_0$ and $\sigma_0$, can be defined in terms of $\overline{LWC}$ and $f_{LWC}$ in the following
fashion:

$$LWC_0 = \frac{\overline{LWC}}{\sqrt{f_{LWC}+1}}, \tag{B2}$$

$$\sigma_0^2 = \ln(f_{LWC}+1). \tag{B3}$$

A gamma distribution of LWC is defined as:

$$p(LWC) = \frac{1}{\Gamma(\nu)}\left(\frac{\nu}{\overline{LWC}}\right)^\nu LWC^{\nu-1} \exp\left[ -\frac{\nu LWC}{\overline{LWC}} \right], \tag{B4}$$





where $\Gamma(\nu)$ denotes the gamma function and the parameter of the distribution $\nu$ is related to $f_{LWC}$ as follows:

$$\nu = \left(\frac{1}{f_{LWC}}\right)^2. \tag{B5}$$

*Author contributions.* NC and BM both contributed to the development of the Tripleclouds radiation solver and its subsequent implementation in *libRadtran*. NC performed the numerical experiments, evaluated the data and interpreted the results together with BM. NC prepared
the manuscript with contributions from BM.

*Competing interests.* The authors declare that they have no conflict of interest.

*Acknowledgements.* The research leading to these results has been done within the subproject "B4: Radiative heating and cooling at cloud scale and its impact on dynamics" of the Transregional Collaborative Research Center SFB / TRR 165 "Waves to Weather" funded by the German Science Foundation (DFG). We thank Dr. Fabian Jakub for providing us with the UCLA-LES cloud field. We further thank Dr.
Audine Laurian and Mihail Manev for their perceptive comments on an earlier version of the manuscript.



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
