# Peer review of "The incorporation of the Tripleclouds concept into the $\delta$ -Eddington two-stream radiation scheme: solver characterization and its application to shallow cumulus clouds"

_Atmospheric Chemistry and Physics, 2020_

## Referee Comment (RC1) · Lazaros Oreopoulos (Referee) · 11 Jun 2020

This paper introduces a new formulation for the triple cloud (TC) scheme which provides a simple framework for dealing with subgrid cloud heterogeneity in GCM radiative transfer calculations. The new formulation has to do with how the thick portion of the cloud in triple clouds (triple = thick cloud, thin cloud, clear) is treated in terms of overlap for adjacent cloud layers. Fig. 9 also nicely summarizes differences in the way

upward reflected fluxes are distributed to the cloud layer above. The authors claim that a new element (even call it "pioneering") of their work is validation using surface fluxes and heating rates, as opposed to previous TOA only flux validation, but this is a trivial consideration, since the previous triple scheme can do that too, it's just never been presented. The basic structure of the paper is that the modified TC scheme is introduced and then applied on a simplified (according to TC rules) version of an LES cumulus field, both for SW (different solar angles) and then LW. The results are compared against the ICA calculation (should be better) and a full Monte Carlo calculation that accounts for full 3D cloud structure (should be even better); also against a standard non-TC GCM scheme (which should be worse). Various choices on how to distribute the cloud layer liquid water content to the thick and thin regions are presented. Results are slightly different between the different TC versions, but almost always better than a traditional non-TC GCM RT scheme, except perhaps for surface LW fluxes (which are though very small in a net flux sense because the upward and downward LW fluxes for a cloud field with low bases are very similar).

Overall, this is a good paper that I enjoyed reading and needs only some minor fixes (per comments below) to see the light of day as a proper ACP paper.

Here are some specific comments:

The assumption that the thicker part of the cloud will be towards the center is reasonable, but demonstrating that with fig. 3 is almost irrelevant because that figure shows the centers of multiple small clouds, which the GCM does not represent. The GCM implicitly only has a single cloud in a 50-100 km grid cell.

It is not clear how clouds overlap is treated when the cloud layers are separated by a clear layer. Is it random overlap then? How would one of the Fig. 8 panels look if there was a single clear layer between the two cloudy layers? So, is the overlap only considered for neighboring cloudy layer pairs? Are pairs of cloudy layers that are distant completely independent even if there is no clear layer in-between? In other words, only

a pair-wise coupling of fluxes is considered? Generalized (exponential-random) overlap can overlap any pair of clouds, but of course explicit radiative treatment is messy (if not impossible) without subcolumns.

I'm not convinced that this method is better than McICA because the latter can operate on any subcolumns which can be generated with more realistic rules for overlap and LWC subgrid variability (and its overlap). I mean, if exponential-random agrees better with observations, why not try to use it? The authors state that the McICA noise may be undesirable and impactful, and that's perhaps true, but perhaps this is less important than achieving smaller systematic biases? I don't buy the argument in lines 72-74 that fewer spectral intervals will make McICA worse. This seems to assume that you have to produce only as many columns as g-points so that each column is paired with one g-point, but this doesn't have to be the case. One can easily generate Nc-multiple of g-point subcolumns (i.e., a total Nc*Ng subcolumns) of so that the same spectral point operates on Nc subcolumns. This will reduce the noise. The fewer g-points, the less the McICA noise, actually.

Line 39: Barker (1996) is not the best reference in this case. That paper deals with horizontal inhomogeneity of single cloud layers, therefore irrelevant for GCMs. A much better reference is Oreopoulos and Barker, QJRMS (1999) which deals with multiple vertically overlapped cloudy layers each of which has a gamma distribution of LWC. That scheme was specifically designed for GCMs and was actually deployed on a couple (no papers exist though), but was quickly superseded by McICA.

Lines 27-29: When reference is made to the maximum-random overlap assumption, everyone assumes that there is a unique implementation, but that's simply not true! Indeed, there are various flavors. Geleyn and Hollingsworth (1979) may actually be the best one. But there is actually the cloud "block" implementation of max-ran, which is visually captured in Fig. 10 of Chou et al., JAS (1998). In GH79, two cloudy layers that have another cloudy layer in-between are still assumed to be maximally overlapped for the common portion they have with the in-between layer, but the portions of the layers

that correspond to the clear fraction of the in-between layer are randomly-overlapped. In the Chou et al. representations these clouds would be maximally overlapped if they belonged to the same high, middle, or low block. Only the blocks themselves are randomly overlapped. But the Chou et al. (1998) is still called a max-ran scheme, yet is it very different than GH79! Also, incidentally, the Morcrette and Fouquart paper does not discuss or advocate for max-ran overlap. Rather, it compares, max, min, and two versions of random overlap.

Line 306, leave it to the reader. Well, it's not very common to ask such a thing! Why don't you include these other three cases in the Appendix?

And a few minor corrections:

Line 61: "pioneering"

Line 79: "exertion"? You mean version?

Line 88: "pairs".

Table 1: Odd to call the third experiment "TSM". All experiments not conducted with MC are TSM. So strictly speaking, you have TSM-ICA, TSM-HOM, and TSM-GCM. You could have also conducted the experiment in the middle field with MC, i.e, MC-HOM. Would still have 3D effects because of finite cloud sizes, but no effects due to internal cloud heterogeneity.

Line 376: "validity"

Lines 378-379: No discussion of the Fig. 12 results?

Line 419: You mean rightmost column?

Line 476: Or there was no bias reduction at all!

Lines 485-486: Another marine BL cloud classification you may want to mention is this: https://www.atmos-meas-tech-discuss.net/amt-2020-61/

---

## Referee Comment (RC2) · James Manners (Referee) · 23 Jun 2020

This paper presents the implementation of the "tripleclouds" treatment of cloud inhomogeneity into the libRadtran radiative transfer package. This includes the development of a new "maximum-random" overlap technique to represent the core-shell model of clouds. A new solver is also developed for the treatment of two-stream fluxes in three regions within the column (although it is unclear from the description whether this

method is novel or essentially a reimplementation of a previously documented method). The new tripleclouds formulation is tested with application to a shallow cumulus cloud field from an LES simulation, by comparison with a full 3D Monte Carlo scheme, an independent column approximation method, and a homogeneous cloud fraction method. This comparison is particularly interesting for having a full 3D model as the control and seeing how the error introduced by the treatment of inhomogeneity compares with the error from neglecting 3D effects.

The paper is generally very clearly written with some useful schematics. I would recommend it for publication subject to some minor revisions commented on below.

Specific comments:

1) Introduction: I would suggest that one disadvantage of the tripleclouds method, compared to the other cloud heterogeneity methods described, is the computational cost of the tripleclouds solver. Lines 72-74 mention that the value of the tripleclouds scheme would be increased if fewer spectral intervals were used. Perhaps the main point to mention here is that in order to limit MCICA noise when there are a small number of spectral intervals, oversampling of each interval would be required, which would increase the cost of MCICA to a similar level as the tripleclouds solver.

2) Lines 77-78: the initial implementation of the "tripleclouds" scheme from Shonk and Hogan 2008 was in the Edwards-Slingo (now "Socrates") model that is also a delta-Eddington two-stream scheme. I would suggest the novel focus of this paper is the implementation and adaptation of the method in the libRadtran package in particular.

3) Section 2.3.2 conventional GCM representation: did you have an optical depth threshold to determine the cloudy part of the domain? Might the results improve if you did? The determination of cloud fraction in a GCM is quite model dependent I imagine and possibly tuned to give the best emergent cloud properties. It probably doesn't represent the total cloud fraction down to the very thinnest cloud.

4) Section 3.1: thermal emission is neglected in these equations and could be simply added as an extra source term in equation 4 and 6, even if it is to be neglected in the further equations.

5) Line 249: As a suggestion, I think the overlap (transfer) coefficients should correspond to a level rather than a layer as they determine the transfer across the boundary between layers. It would then be useful to add the level being referred to for each T in equations 10, 11, and 12. Note then that eg. $T\_up^{ck,cn}(i) = T\_down^{cn,ck}(i)$, so the up and down arrows are perhaps redundant and the notation could simply indicate the upper cloud region, lower cloud region.

6) Section 3.3: While the formulation of the overlap rules is fairly clearly outlined here I think it would be better to provide the generalised formulas for the overlap between different regions rather than just the example case given. Especially as I think this method might be one of the key novel developments in this scheme. It would be particularly interesting to see how this new overlap scheme performs in comparison to a standard maximum-random approach which does not follow a core-shell model (i.e. a scheme where each region is maximally overlapped with itself but the overhang randomly overlapped with the other regions).

7) Section 3.4: I think this section requires further explanation with regard to how exactly your solver is implemented. Ideally, this should be explained in relation to the concept of entrapment explained in Hogan et al 2019. The method implemented in Shonk and Hogan 2008 corresponds to zero entrapment whereas the original Edwards and Slingo / Socrates method described in eqn 15 of Shonk and Hogan 2008 corresponds to maximum entrapment. It looks to me like your method also corresponds to maximum entrapment. It would be useful to indicate how your method differs from this.

8) Figure 9: This schematic is not entirely clear: I think the large downward radiation arrow should actually indicate the flux coming from just the upper dark blue region.

9) Section 5.1: At large zenith angles your TC schemes tend to approximate the 3D

heating better than the ICA: could this be due to your effective treatment of "maximum entrapment" in your TC solver, whereas the ICA effectively treats "zero entrapment" (from Hogan et al. 2019)? The effective treatment of 3D effects in your method should be discussed, otherwise the improved treatment of TC over ICA can only be interpreted as a cancellation of errors.

10) Section 5.1: The use of a constant FSD of 0.75 in these experiments muddies the comparison a bit as you are convoluting the error in using the constant FSD with the error introduced by the method to generate the LWC pair. You could repeat the experiments using the actual FSD in each layer to isolate error in the LWC pair method.

11) Section 5.2: The performance of the TC scheme for surface thermal flux should probably be compared with the ICA as the achievable benchmark as the entrapment implicit in your scheme would not have a large effect in the thermal and you scheme is effectively approximating the ICA.

12) Appendix A: this looks like something that would be better left to a user manual rather than a journal paper - with development of the package I suspect these instructions would change and the user manual could be updated accordingly.

Technical corrections:

1) Line 184: stemms -> stems

2) Line 434: Hill 2015 is referenced but is not in the reference list (Hill et al 2015: A regime-dependent parametrization of subgrid-scale cloud water content variability). This paper could also be referenced at line 480/481 in the conclusions.
* * *

---

## Author Comment (AC1) · 12 Jul 2020

**Authors' reply to comments by referee #1 (Lazaros Oreopoulos)**

We thank the referee for carefully reviewing the manuscript, providing the valuable comments and suggestions, which helped improving the original manuscript version. We incorporated the vast majority of suggested improvements. In the following the referee comments are presented in blue and the authors' reply in black. Please note that the line and figure numbers refer to those in the original manuscript. Changes in the revised manuscript are marked with quotation marks and additional indent.

Here are some specific comments:

The assumption that the thicker part of the cloud will be towards the center is reasonable, but demonstrating that with fig. 3 is almost irrelevant because that figure shows the centers of multiple small clouds, which the GCM does not represent. The GCM implicitly only has a single cloud in a 50-100 km grid cell.

It is true that the GCM only has a single cloud in a grid cell, but in reality such cloudy layer would mostly consist of several clouds. Fig. 3 therefore shows the realistic shallow cumulus cloud field consisting of multiple clouds, which would all be subgrid clouds from the GCM point of view. Each of these clouds conforms to the core-shell model, where the optically thicker part is located in cloud interior. The averaged effect should be captured in a GCM. (See also line 103: "Clouds in a cloud field have multiple cores, whereby their aggregate effect can be modelled with a core-shell model.")

It is not clear how clouds overlap is treated when the cloud layers are separated by a clear layer. Is it random overlap then? How would one of the Fig. 8 panels look if there was a single clear layer between the two cloudy layers? So, is the overlap only considered for neighboring cloudy layer pairs? Are pairs of cloudy layers that are distant completely independent even if there is no clear layer in-between? In other words, only pairwise coupling of fluxes is considered? Generalized (exponential-random) overlap can overlap any pair of clouds, but of course explicit radiative treatment is messy (if not impossible) without subcolumns.

Yes, if cloudy layers are separated by a clear layer, they overlap randomly. This is stated in lines 273-275: "We apply the widely used maximum-random overlap assumption (Geleyn and Hollingsworth, 1979) for the entire layer cloudiness (sum of optically thick and thin cloudy regions), where adjacent cloudy layers exhibit maximal overlap and cloudy layers separated by at least one cloud-free layer exhibit random overlap." Correct, we have only considered pairwise overlap. This is expressed in line 309: "Pairwise overlap as employed here ensures that the matrix problem is fast to solve." So the maximum overlap is applied in pairs of adjacent layers for the entire layer cloudiness as well as additionally for the optically thicker part. In order to further emphasize and clarify the latter issue we changed the sentence in line 277 to:

> "We additionally assume the maximum overlap of optically thicker cloudy regions in pairs of adjacent layers."

Similarly, as for the entire layer cloudiness, the random overlap is automatically fulfilled also for the optically thicker cloudy regions separated by at least one cloud-free layer.

I'm not convinced that this method is better than McICA because the latter can operate on any subcolumns which can be generated with more realistic rules for overlap and LWC subgrid variability (and its overlap). I mean, if exponential-random agrees better with observations, why not try to use it? The authors state that the McICA noise may be undesirable and impactful, and that's perhaps true, but perhaps this is less important than achieving smaller systematic biases? I don't buy the argument in lines 72-74 that fewer spectral intervals will make McICA worse. This seems to assume that you have to produce only as many columns as g-points so that each column is paired with one g-point, but this doesn't have to be the case. One can easily generate $N_c$-multiple of g-point subcolumns (i.e., a total $N_c \cdot N_g$ subcolumns) of so that the same spectral point operates on $N_c$ subcolumns. This will reduce the noise. The fewer g-points, the less the McICA noise, actually.

We are not saying that the TC is better than the McICA, rather a possible alternative, which however requires further evaluation. You are right – whereas our current overlap formulation should be well suited for the present shallow cumulus case, it is inadequate for vertically developed cloud systems in strongly sheared

conditions. Therefore we plan to generalize the overlap rules in the next step. As explained in the text the impact of the McICA noise can be harmful (inducing undesired feedback loops) – for example for low clouds, which are essentially maintained by local cloud top radiative cooling. Thereby the TC might be a better option, eventually also leading to smaller systematic errors in such critical cases. We have however additionally emphasized that the McICA is computationally faster than the TC. The argument about fewer spectral intervals worsening the performance of the McICA is summarized after Hogan and Bozzo (2016) as stated in the sentence. We have however improved this part as suggested also by referee #2:

> "In contrast to the McICA, which is still operational also at EMCWF due to ist higher computational efficiency, the TC scheme does not produce any radiative noise. As suggested by Hogan and Bozzo (2016) this superiority could become even more valuable in the future if an alternative gas optics model with fewer spectral intervals than the current RRTM-G (Mlawer et al., 1997) will be developed, since this would increase the level of the McICA noise, but it would not affect the Tripleclouds. In other words, in order to limit the McICA noise in this case, oversampling of each interval would be required, which could increase the computational cost of the McICA to a similar degree as that of the Tripleclouds scheme."

The fair comparison of the McICA and the TC is beyond the scope of this study, but is should be carried out in the next step.

Line 39:  Barker (1996) is not the best reference in this case.  That paper deals with horizontal inhomogeneity of single cloud layers, therefore irrelevant for GCMs. A much better reference is Oreopoulos and Barker, QJRMS (1999) which deals with multiple vertically overlapped cloudy layers each of which has a gamma distribution of LWC. That scheme was specifically designed for GCMs and was actually deployed on a couple (no papers exist though), but was quickly superseded by McICA.

Thank you for this suggestion. We changed the reference to Oreopoulos and Barker (1999).

Lines 27-29:  When reference is made to the maximum-random overlap assumption, everyone assumes that there is a unique implementation, but that's simply not true! Indeed, there are various flavors. Geleyn and Hollingsworth (1979) may actually be the best one.  But there is actually the cloud "block" implementation of max-ran, which is visually captured in Fig. 10 of Chou et al., JAS (1998). In GH79, two cloudy layers that have another cloudy layer in-between are still assumed to be maximally overlapped for the common portion they have with the in-between layer, but the portions of the layers that correspond to the clear fraction of the in-between layer are randomly-overlapped. In the Chou et al.  representations these clouds would be maximally  overlapped  if they belonged to the same high, middle, or low block. Only the blocks themselves are randomly overlapped. But the Chou et al. (1998) is still called a max-ran scheme, yet is it very different than GH79!  Also, incidentally, the Morcrette and Fouquart paper does not discuss or advocate for max-ran overlap.  Rather, it compares, max, min, and two versions of random overlap.

We kept the original reference of Geleyn and Hollingsworth (1979), as we also think it is the most appropriate in this case. Furthermore, we removed the reference of Morcrette and Fouquart (1986) in the context of advocating the maximum-random overlap.

Line 306, leave it to the reader. Well, it's not very common to ask such a thing!  Why don't you include these other three cases in the Appendix?

Thank you for pointing this out. We added an extra Appendix section and changed the sentence to:

> "The derivation of overlap coefficients for other three geometries involves analogous considerations, whereby the resulting formulas as well as their generalized formulation are given in Appendix B."

And a few minor corrections:

Line 61: "pioneering"

We agree that "pioneering" is not the correct term. We changed the sentence to: "In the primary work of SH08 ..." in line 61. We further removed the word "pioneeringly" from line 16.

Apparently the "exertion" was not the best wording. We changed it to "incorporation", which also makes this sentence consistent with the paper title.

Changed.

Table 1: Odd to call the third experiment "TSM". All experiments not conducted with MC are TSM. So strictly speaking, you have TSM-ICA, TSM-HOM, and TSM-GCM. You could have also conducted the experiment in the middle field with MC, i.e, MC-HOM. Would still have 3D effects because of finite cloud sizes, but no effects due to internal cloud heterogeneity.

You are right, these experiments could be named "TSM-ICA", "TSM-HOM" and "TSM-GCM", but we prefer to name them as short as possible, assuming it is clear they have all been performed with the two-stream method as described in the text. Therefore the first one of the three TSM experiments is simply called "TSM" (to distinguish it from the Monte Carlo ICA experiment, which is termed "ICA"), whereas other TSM experiments are abbreviated to "HOM" and "GCM". Yes, we actually conducted the MC experiment on the cloud field with removed horizontal heterogeneity as well, but it is not presented in the paper, since it does not bring any other conclusions.

Changed.

Thank you for this suggestion, we extended the paragraph as follows:

> "Further, to test the sensitivity of TC radiative quantities to the assumed form of the subgrid cloud condensate distribution, we employed the FSD method in conjunction with all three distribution assumptions (Gaussian, gamma, lognormal). The resulting LWC profiles are shown in Fig. 12, demonstrating that the LWC pair characterizing the two cloudy regions is clearly sensitive to the distribution assumption, when mean global FSD estimate is used as a proxy for cloud horizontal inhomogeneity degree."

We simplified this parenthesis to contain only the figure number, as one should actually compare the middle and rightmost column.

We changed the sentence part to: "... the nighttime bias was slightly enlarged, ...".

Thank you for providing this reference, which we included in the text:

> "The classification of rich spatial patterns into various mesoscale cloud morphologies can thereby valuably be performed with deep learning algorithms (e.g., Yuan et al., 2020)."

---

## Author Comment (AC2) · 12 Jul 2020

**Authors' reply to comments by referee #2 (James Manners)**

We thank the referee for carefully reviewing the manuscript, providing the valuable comments and suggestions, which helped improving the original manuscript version. We incorporated the majority of suggested improvements. In the following the referee comments are presented in blue and the authors' reply in black. Please note that the line and figure numbers refer to those in the original manuscript. Changes in the revised manuscript are marked with quotation marks and additional indent.

Specific comments:

1) Introduction: I would suggest that one disadvantage of the tripleclouds method, compared to the other cloud heterogeneity methods described, is the computational cost of the tripleclouds solver. Lines 72-74 mention that the value of the tripleclouds scheme would be increased if fewer spectral intervals were used. Perhaps the main point to mention here is that in order to limit MCICA noise when there are a small number of spectral intervals, oversampling of each interval would be required, which would increase the cost of MCICA to a similar level as the tripleclouds solver.

Thank you for this advice. We extended the relevant paragraph as you suggested and additionally emphasized that the current operational McICA is computationally more efficient than the Tripleclouds:

> "In contrast to the McICA, which is still operational also at EMCWF due to ist higher computational efficiency, the TC scheme does not produce any radiative noise. As suggested by Hogan and Bozzo (2016) this superiority could become even more valuable in the future if an alternative gas optics model with fewer spectral intervals than the current RRTM-G (Mlawer et al., 1997) will be developed, since this would increase the level of the McICA noise, but it would not affect the Tripleclouds. In other words, in order to limit the McICA noise in this case, oversampling of each interval would be required, which could increase the computational cost of the McICA to a similar degree as that of the Tripleclouds scheme."

2) Lines 77-78: the initial implementation of the "tripleclouds" scheme from Shonk and Hogan 2008 was in the Edwards-Slingo (now "Socrates") model that is also a delta-Eddington two-stream scheme. I would suggest the novel focus of this paper is the implementation and adaptation of the method in the libRadtran package in particular.

Thank you for pointing this out. We changed the text accordingly:

> "To that end, building upon the Tripleclouds idea of SH08, the classic δ-Eddington two-stream method with maximum-random overlap assumption for partial cloudiness was extended to incorporate an extra cloudy region at each height (Fig. 1, bottom right). The prime focus of this paper is to document the present Tripleclouds implementation in the comprehensive radiative transfer package libRadtran (Mayer and Kylling, 2005; Emde et al., 2016)."

3) Section 2.3.2 conventional GCM representation: did you have an optical depth threshold to determine the cloudy part of the domain? Might the results improve if you did? The determination of cloud fraction in a GCM is quite model dependent I imagine and possibly tuned to give the best emergent cloud properties. It probably doesn't represent the total cloud fraction down to the very thinnest cloud.

Yes, we applied a standard LWC threshold of $10^{-3}$ g/m$^3$ to define a cloudy pixel on the LES grid. This should give reasonable LES cloud representation as well as reasonable derived GCM cloudiness, and consequently also the heating rate.

4) Section 3.1: thermal emission is neglected in these equations and could be simply added as an extra source term in equation 4 and 6, even if it is to be neglected in the further equations.

Thank you for this suggestion. We added an extra paragraph within Section 3.1 briefly explaining the thermal emission treatment. As our current version of the two-stream radiation scheme is only capable of separately

performing the solar and thermal calculations, we prefer not to simultaneously include the thermal emission term in Eqs. (4) and (6). The added paragraph is the following:

> "The preceding formulation considered solar radiative transfer in the absence of thermal emission. As solar and thermal spectra are separated and can be therefore conveniently treated independently, the solar source is merely replaced with the terrestrial emission term when addressing thermal radiation. The vertical temperature variation is thereby taken into account by allowing the Planck function to vary in accordance with the Eddington type linearization: $B_{Planck}(\tau) = B_0 + B_1 \hat{} \tau$, where $B_0$ and $B_1$ are constants. The equation system for a single layer is constructed in a similar manner, accounting for both upward and downward thermal emission contributions. For a more comprehensive explanation the reader is referred to Zdunkowski et al. (2007), as in the rest of this section we will focus on solar radiation."

5) Line 249: As a suggestion, I think the overlap (transfer) coefficients should correspond to a level rather than a layer as they determine the transfer across the boundary between layers. It would then be useful to add the level being referred to for each T in equations 10, 11, and 12. Note then that eg. T_up^ck,cn(i) = T_down^cn,ck(i), so the up and down arrows are perhaps redundant and the notation could simply indicate the upper cloud region, lower cloud region.

The overlap coefficients could be expressed as level quantities and hence presumably without distinguishment between up and down arrows. For consistency, however, we would like to preserve the same indexing in the paper as in our coded Tripleclouds implementation, where the overlap coefficients are defined per layer (this is further consistent with our recently implemented "twomaxrnd" solver following Zdunkowski et al., 2007). We have further emphasized this in the text:

> "The coefficients starting with T appearing in Eqs. 10, 11, 12 are referred to as the overlap (transfer) coefficients and correspond to the layer under consideration (j)."

As they all correspond to the same layer (j) we omitted this in Eqs. 10, 11, 12 - consistently with the omission of the j-index for the Eddington coefficients. In this case the upward and downward arrows are necessary in Eqs. 10, 11, 12, since T_down^a,b(j) = function(C(j),C(j-1)) and T_up^a,b(j) = function(C(j),C(j+1)). We have further emphasized the latter:

> "The transmission of upward radiation is managed via overlap coefficients T_up^{a,b}(j) in an equivalent manner, except that these are dependent on the cloud fraction in the layer under consideration and that in the layer underneath [C(j), C(j+1)]."

6) Section 3.3: While the formulation of the overlap rules is fairly clearly outlined here I think it would be better to provide the generalised formulas for the overlap between different regions rather than just the example case given. Especially as I think this method might be one of the key novel developments in this scheme. It would be particularly interesting to see how this new overlap scheme performs in comparison to a standard maximum-random approach which does not follow a core-shell model (i.e. a scheme where each region is maximally overlapped with itself but the overhang randomly overlapped with the other regions).

As the referee #1 also suggested that the initial description of overlap rules including only one cloud geometry case is not sufficient, we added an extra overlap section in the Appendix. This section contains the overlap coefficients for the four possible geometries as well as their generalized formulas. We agree that comparison of this overlap scheme with the standard maximum-random approach for three regions would be interesting, but it is out of the scope of the present study.

7) Section 3.4: I think this section requires further explanation with regard to how exactly your solver is implemented. Ideally, this should be explained in relation to the concept of entrapment explained in Hogan et al 2019. The method implemented in Shonk and Hogan 2008 corresponds to zero entrapment whereas the original Edwards and Slingo / Socrates method described in eqn 15 of Shonk and Hogan 2008 corresponds to maximum entrapment. It looks to me like your method also corresponds to maximum entrapment. It would be useful to indicate how your method differs from this.

Thank you for this suggestion, we agree that Section 3.4 was not adequately formulated. From the various entrapment possibilities presented in Hogan et al. (2019) ["zero", "explicit" and "maximum" entrapment; their Fig. 1] it might seem that our version corresponds best with the maximum entrapment. Nevertheless, Fig. 1 of Hogan et al. (2019) illustrates the "entrapment" as a mechanism occurring between two randomly overlapped layers of a multilayered cloud scene, whereas our Fig. 9 (right panel, present implementation) illustrates the division of radiative fluxes between two adjacent maximally overlapped cloudy layers. This division is managed according to the assumed overlap: whereas our overlap treatment follows the core-shell model, their does not. The exact comparison of both solvers (in theory and in practice) should be a topic of a future study. We therefore removed Section 3.4 from the current version and rather briefly clarified the differences in the initial introductory part of Section 3:

> "The underlying δ-Eddington two-stream framework employed in the present Tripleclouds implementation differs from that applied by SH08 and subsequent studies (e.g., Shonk et al, 2010; Hogan et al., 2019), whereby the latter is based on the Adding Method (Lacis and Hansen, 1974) as originally included in the Edwards and Slingo (1996) radiation scheme. Therefore we first present the δ-Eddington two-stream method (Zdunkowski et al., 2007), already previously contained in *libRadtran*, and introduce the terminology in Sect. 3.1. We focus only on those aspects of the method, important to understand its extension to multiple (three) regions, explained in subsequent Sect. 3.2. The novel overlap formulation based on the core-shell model is established in Sect. 3.3. Further technical instructions regarding the Tripleclouds usage within the scope of *libRadtran* are provided in Appendix A."

8) Figure 9: This schematic is not entirely clear: I think the large downward radiation arrow should actually indicate the flux coming from just the upper dark blue region.

We removed Section 3.4 and thereby this figure in the revised version, therefore the details might not be relevant anymore. Nevertheless, in our Tripleclouds implementation the large downward arrow represents the entire downward radiative flux that is entering the region of optically thick cloud in the layer (j) under consideration. This flux component stems from all three regions in the upper layer and not only from the optically thick cloudy region.

9) Section 5.1: At large zenith angles your TC schemes tend to approximate the 3D heating better than the ICA: could this be due to your effective treatment of "maximum entrapment" in your TC solver, whereas the ICA effectively treats "zero entrapment" (from Hogan et al. 2019)? The effective treatment of 3D effects in your method should be discussed, otherwise the improved treatment of TC over ICA can only be interpreted as a cancellation of errors.

This is indeed an interesting note. We extended the discussion within Section 5.1 accordingly:

> "Finally, it should be noted that at low Sun (SZA of 30° and 60°) the TC is generally even more accurate than the ICA, which could be partially due to effective treatment of solar 3-D effects in the TC scheme."

We as well added an extra sentence comparing the TC and the ICA in the thermal spectral range:

> "Noteworthy, the TC performs similarly well as the ICA also in the thermal spectral range, implying that the realistic subgrid cloud variability can be adequately represented by a two-point PDF."

10) Section 5.1: The use of a constant FSD of 0.75 in these experiments muddies the comparison a bit as you are convoluting the error in using the constant FSD with the error introduced by the method to generate the LWC pair. You could repeat the experiments using the actual FSD in each layer to isolate error in the LWC pair method.

We repeated the experiments using the actual FSD in each layer as you suggested. We additionally repeated the experiments with the parameterization of Boutle et al. (2014) for liquid cloud inhomogeneity. We have eventually decided to include the results of the latter, which is of practical interest for the application in weather and climate models, pointing out limitations of current FSD parameterizations. We added an extra

figure panel within Section 5.1 and extended the corresponding discussion in Section 5.1 and 5.2 as well as slightly changed the summary and conclusions in Section 6.

The added paragraph in Section 5.1:

> "Based upon these considerations, we additionally evaluated the parameterization of Boutle et al. (2014) for liquid cloud inhomogeneity, which takes into account that variability is cloud fraction dependent. Although solar RMSE slightly reduces when FSD is represented following Boutle et al. (2014), the TC experiment with global FSD constant assuming Gaussian distribution remains the most accurate during both nighttime and daytime (Fig. 13, right). To that end, the development of improved parameterizations is highly desired in the future."

The added comment in Section 5.2:

> "Similar findings are obtained if the FSD is parameterized according to Boutle et al. (2014), which does not bring desired improvements (not shown)."

The changed sentence in Section 6:

> "The validity of global estimate of fractional standard deviation (a common measure of cloud horizontal variability) as well as of more sophisticated inhomogeneity parameterization was tested along with different assumptions for subgrid cloud condensate distribution (Gaussian, gamma, lognormal), which are frequently applied when representing clouds in weather and climate models."

11) Section 5.2: The performance of the TC scheme for surface thermal flux should probably be compared with the ICA as the achievable benchmark as the entrapment implicit in your scheme would not have a large effect in the thermal and you scheme is effectively approximating the ICA.

The performance of the Tripleclouds should always preferably be compared with the 3-D calculation as a benchmark.

12) Appendix A: this looks like something that would be better left to a user manual rather than a journal paper - with development of the package I suspect these instructions would change and the user manual could be updated accordingly.

We shortened the appendix by removing the instructions for "twomaxrnd" solver, which is not the main focus of this paper. We however kept the Tripleclouds instructions in order to additionally highlight the simple usage of the solver. Otherwise yes – similar guidance will be provided in the user manual accompanying the next libRadtran release.

Technical corrections:

1) Line 184: stemms -> stems

Changed.

2) Line 434:  Hill 2015 is referenced but is not in the reference list (Hill et al 2015: A regime-dependent parametrization of subgrid-scale cloud water content variability). This paper could also be referenced at line 480/481 in the conclusions.

Corrected, we included Hill et al. (2015) in the reference list. We also added this reference within the conclusions section, together with similar studies of Hill et al. (2012) and Boutle et al. (2014).

*Additional remark: We have further added a brief preface at the beginning of Section 2 (introducing subsections 2.1-2.3; to make it consistent with prefaces in Sections 3, 4, 5):*

> *"We first introduce the core-shell model for convective clouds as well as the shallow cumulus case study in Sect. 2.1. The radiative transfer models and experimental setup are outlined in Sect. 2.2. The results of preliminary radiation experiments demonstrating the importance of representing cloud horizontal heterogeneity are presented in Sect. 2.3."*

*Consequently, we could shorten/reformulate the last paragraph of the Introduction as follows:*

> *"The manuscript is organized as follows: in Sect. 2 the cloud data and methodology is introduced. In Sect. 3 our version of the TC radiation scheme is presented. In Sect. 4 existing approaches for generating cloud condensate pairs are revised. The TC performance is evaluated in Sect. 5. A brief summary and concluding remarks are given in Sect. 6."*

---

## Author Response (AR2)

**Authors' reply to Editor about final technical corrections**

Dear Editor,

thank you for your positive reply. We are pleased that our manuscript is accepted for publication in ACP.

We have addressed the suggested technical corrections as follows:

1.) We have added the information about the applied LWC threshold within Section 2.1.2.

2.) We have clarified Fig. 3: The cloud mask panel is retained, whereas the vertically integrated cloud optical thickness plot is provided only once (replotted in grey shading and equipped with colorbar, so there is no ambiguity about thresholds anymore). Consequently, we removed the middle panel of Fig. 2 as well, since it is a similar visualization (only using a different colormap).

3.) In accordance with ACP manuscript preparation quidelines using LaTeX, we did not include extra commands and packages in the *.tex file (which we did in the initial version). Therefore we slightly modified the appendix formatting - we introduced Fig. A1 and removed colors in Table B1.

Best wishes
Nina Črnivec on behalf of both authors